# Late summer transition from a free-tropospheric to boundary layer source of Aitken mode aerosol in the high Arctic

Ruth Price[1], Andrea Baccarini[2,*], Julia Schmale[2], Paul Zieger[3,4], Ian M. Brooks[1], Paul Field[1,5], and Ken S. Carslaw[1]

[1]School of Earth and Environment, University of Leeds, Leeds, UK
[2]Extreme Environments Research Laboratory, École Polytechnique Fédérale de Lausanne, Sion, Switzerland
[3]Department of Environmental Science, Stockholm University, Stockholm, Sweden
[4]Bolin Centre for Climate Research, Stockholm University, Stockholm, Sweden
[5]Met Office, Exeter, UK
[*]now at: Laboratory of Atmospheric Processes and their Impacts, École Polytechnique Fédérale de Lausanne, Lausanne, Switzerland

**Correspondence:** Ruth Price (eersp@leeds.ac.uk)

**Abstract.** In the Arctic, the aerosol budget plays a particular role in determining the behaviour of clouds, which are important for the surface energy balance and thus for the region's climate. A key question is the extent to which cloud condensation nuclei in the high Arctic summertime boundary layer are controlled by local emission and formation processes as opposed to transport from outside. Each of these sources is likely to respond differently to future changes in ice cover. Here we use a global model and observations from ship and aircraft field campaigns to understand the source of high Arctic aerosol in late summer. We find that particles formed remotely, i.e., at latitudes outside the Arctic, are the dominant source of boundary layer Aitken mode particles during the sea ice melt period up to the end of August. Particles from such remote sources, entrained into the boundary layer from the free troposphere, account for nucleation and Aitken mode particle concentrations that are otherwise underestimated by the model. This source from outside the high Arctic declines as photochemical rates decrease towards the end of summer, and is largely replaced by local new particle formation driven by iodic acid created during freeze-up. Such a local source increases the simulated Aitken mode particle concentrations by two orders of magnitude during sea ice freeze-up and is consistent with strong fluctuations in nucleation mode concentrations that occur in September. Our results suggest a high-Arctic aerosol regime shift in late summer, and only after this shift do cloud condensation nuclei become sensitive to local aerosol processes.

## 1 Introduction

The Arctic is a key component of the global climate system. Over the past five decades, the mean surface temperature of the Arctic has increased 3–4 times faster than the global average (AMAP, 2021; Rantanen et al., 2022). Regional changes in the Arctic can have global impacts, such as climate feedbacks from albedo changes due to loss of sea ice, land ice and snow (Notz and Stroeve, 2018; Zhang et al., 2019; Flanner et al., 2011); changes in carbon sources due to increased wildfires (Walker et al.,

2019; Randerson et al., 2006) and melting permafrost (Hugelius et al., 2014; Biskaborn et al., 2019); and global sea level rise from melting of the Greenland ice sheet (Cazenave et al., 2018; Shepherd et al., 2012).

Clouds are a major control on the surface energy balance in the Arctic. Due to the low solar insolation and high albedo of sea ice in the high Arctic (i.e. pack ice regions north of 80 °), the shortwave cooling effect of clouds is less important than at lower latitudes. Instead, longwave effects dominate such that the net radiative effect of low-level Arctic clouds is surface warming for
all but a few weeks in the middle of summer (Curry et al., 1993; Shupe and Intrieri, 2004; Sedlar et al., 2011; Kay et al., 2016). Low clouds are common and can persist in the mixed-phase for several days (Shupe, 2011). Aerosol particles act as cloud condensation nuclei (CCN) and ice-nucleating particles (INP) and therefore influence the behaviour of clouds. Observational Arctic case studies have shown that perturbations to aerosol concentrations can change the radiative effect of clouds (Lubin and Vogelmann, 2006; Garrett and Zhao, 2006; Mauritsen et al., 2011). Modelling studies reproduce this behaviour, while also
highlighting the difficulty in creating models that can accurately simulate the complex behaviour of Arctic clouds (Alterskjær et al., 2010; Birch et al., 2012; Stevens et al., 2018).

The aerosol budget of the high Arctic is a balance of different processes, including primary emissions, new particle formation, condensational growth from vapours, long-range transport of anthropogenic emissions, and wet scavenging (Willis et al., 2018; Schmale et al., 2021). Transport of aerosol from lower latitudes is more efficient during the winter and spring because it
is thermodynamically easier in winter for air to enter the Arctic region from the south than it is in summer (Stohl, 2006). Cold air masses sitting over ice-covered portions of the Arctic Ocean create sharp north-south temperature gradients, described as the polar dome, that act as a barrier for air moving towards the Arctic lower troposphere from the south. In the winter, several factors make it easier for air from south to penetrate the polar dome, including a more southward extent of the dome, and cooling of air during transport due to proximity to snow and ice surfaces over land. Also, the removal of aerosol by frozen
precipitation in winter is less effective than removal by drizzle in summer (Browse et al., 2012; Korhonen et al., 2008b). These seasonal cycles of aerosol transport and removal efficiency results in a transition from Arctic haze in the spring to more pristine conditions in the summer with fewer accumulation mode particles, less anthropogenic influence, and more, smaller nucleation and Aitken mode particles. Such a transition is evident in measurements e.g. from Svalbard (Ström et al., 2003; Engvall et al., 2008; Tunved et al., 2013; Karl et al., 2019), from pan-Arctic observatories (Schmale et al., 2022), or from the recent year-long
MOSAiC campaign in the pack ice region (Boyer et al., 2023).

Unlike the winter, when long-range transport is the dominant source of Arctic aerosols, summertime particles are thought to be strongly controlled by new particle formation (NPF) and growth from precursor vapours, which takes place in many Arctic locations. However, the large variety of vapours that can play a role, as well as the strong seasonal variation in key processes, have made it difficult to understand the main drivers of Arctic NPF. Observations of Arctic NPF are discussed
in Schmale and Baccarini (2021) and will be briefly summarised here. Sulphuric acid ($H_2SO_4$) and methanesulphonic acid (MSA) have been shown to drive NPF and growth in regions close to or influenced by open water such as Svalbard and the Canadian Arctic archipelago (Heintzenberg et al., 2017; Beck et al., 2020; Chang et al., 2011; Willis et al., 2017). The marine biogenic precursor dimethylsulphide (DMS) is the main source of $H_2SO_4$ and MSA to these regions. Open water is also a source of organic vapours which are observed to contribute to condensational growth of small particles (Willis et al., 2017).

Ammonia has biogenic sources in the Arctic, for example seabird colonies, and has been observed to contribute to NPF events in Svalbard and Greenland (Croft et al., 2016; Beck et al., 2020). Iodine is known to be emitted from kelp in coastal areas and iodine-containing compounds have been observed to drive NPF events near the coast of Greenland (Sipil a et al., 2016; Allan et al., 2015). Of relevance to this study, Baccarini et al. (2020a) recently observed NPF driven by iodic acid ($HIO_3$) in the pack ice region of the Arctic Ocean during the sea ice freeze-up, suggesting an iodine source from snow, ice or ocean water. Recent

laboratory results elucidate the chemical pathway for the creation of iodic acid from iodine (Finkenzeller et al., 2022).

Modelling studies have demonstrated the importance of NPF for the budget of Arctic aerosol. Merikanto et al. (2009) and Gordon et al. (2017) used the global aerosol model GLOMAP to show the importance of NPF on a global scale. Both studies indicate that a high fraction (greater than 80%) of particles and CCN in the Arctic are derived from new particle formation. While the model configuration in Merikanto et al. (2009) only included paremeterisations based on $H_2SO_4$, the

model used by Gordon et al. (2017) included parameterisations for neutral and ion-induced binary ($H_2SO_4$-water) and ternary ($H_2SO_4$-ammonia-water) NPF, NPF from organic molecules and $H_2SO_4$, and pure organic NPF driven by highly oxygenated molecules (HOMs). The authors found that a significant fraction (greater than 40%) of Arctic CCN originate from secondary organic aerosol, including HOMs.Korhonen et al. (2008b) and Browse et al. (2014), both using GLOMAP to investigate Arctic aerosol, found that boundary layer NPF driven by $H_2SO_4$ was required to explain measured size distributions or CCN

concentrations in the high Arctic during summer. Karl et al. (2012) used an aerosol dynamics model to study NPF events that were observed during the summers of 1996, 2001 and 2008. $H_2SO_4$ and organic vapours were used in the model to drive NPF events. Simulations of NPF driven by $H_2SO_4$ followed by growth from condensation of organic vapours were able to reproduce the particle size distributions observed during NPF events. The inclusion of organic vapours as a driver of NPF led to an overprediction in the concentration of particles, though the authors note significant uncertainty in the sources and

concentrations of organic vapours in the high Arctic. Croft et al. (2019) used a chemical transport and aerosol microphysics model to show that condensation of secondary organic vapour played a key role in particle formation events in the Canadian Arctic during the summer. The vapours were assumed to have a marine, biogenic source.

Boundary layer NPF as a source of Arctic aerosol implies a potentially high sensitivity of the local aerosol budget to the changing climate. Any future increase in the extent of open water and the marginal ice zone in summer will affect the emission

of aerosol and precursor gases to the atmosphere. Dall'Osto et al. (2017, 2018) have found a correlation between frequency of NPF events and the extent of open water near Svalbard and Greenland, respectively. Such an increase in occurrence of NPF with sea ice loss could be expected to increase CCN concentrations in the Arctic under future warmer conditions, though this is far from certain since changing sea ice extent is not the only controlling factor. Gilgen et al. (2018) found that reduction of sea ice in the year 2050 leads to increased emissions of DMS in a global aerosol-climate model. The increased DMS emissions, along

with increased sea spray aerosol and meteorological changes, cause higher cloud drop number concentrations over the Arctic Ocean. This is in line with results from another global model study, Struthers et al. (2011), which used an atmospheric climate model to investigate the response of sea spray aerosol to sea ice loss. They found a strong increase in sea salt emissions and thus higher cloud drop concentrations, but the effect on clouds and energy budget was uncertain due to poor model representations of aerosol-cloud interactions. In contrast to these studies, the results from Browse et al. (2014) suggest that interactions between

aerosol particles of different sizes could lead to a suppression of NPF in an ice-free Arctic summer. Their results showed an increase in the sink of condensable vapours due to stronger emission of sea spray aerosol from the open water, resulting in a decrease in the concentration of smaller particles from NPF. In addition, the growth of sea spray particles from condensation of vapours shifted the size distribution to larger sizes, leading to enhanced scavenging by precipitation and a decrease in drop concentrations.

The conflicting results from Browse et al. (2014), Gilgen et al. (2018) and Struthers et al. (2011) highlight the difficulty in modelling the Arctic aerosol budget and how it might change in future. Uncertainties in model parameterisations stem from the knowledge gaps in processes controlling the aerosol budget, and cause differences in climate projections from different model set-ups. For example, the sea spray parameterisations used in the Gilgen et al. (2018) and Struthers et al. (2011) studies include empirical representations of the effect of temperature on the sea spray aerosol size distribution, while the parameterisation used in the Browse et al. (2014) study does not. This could be a cause of the discrepancy in their predictions of sea spray aerosol response to sea ice loss, though this has not been studied. The studies also differ in their treatment of primary marine organic emissions, with only the Browse et al. (2012) study including such a source. Some field studies from the high Arctic pack ice region have indicated the importance of primary marine organics in the region (Bigg et al., 2001; Leck and Bigg, 2005; Bigg and Leck, 2008; Leck and Bigg, 2010; Orellana et al., 2011; Karl et al., 2013; Hamacher-Barth et al., 2016), as well as raising questions about possible recycling mechanisms of particles after they are emitted (i.e. through ageing or the particle break-up theory, Leck and Bigg, 1999, 2010; Lawler et al., 2021). The open questions surrounding primary marine emissions complicate modelling of Arctic NPF due to the the sink of condensable vapours from larger particles. Models with size-resolved aerosol microphysics and chemistry are better equipped to study these questions than bulk, single-moment models considering total mass only.

Aitken mode particles can act as CCN in the Arctic, which increases the influence of NPF over the Arctic aerosol budget. Karlsson et al. (2022) used measurements of cloud residual particles (i.e. particles obtained by drying cloud droplets or ice crystals) from the high Arctic to show that Aitken mode particles were acting as CCN during a period of frequent boundary layer new particle formation. Observations of aerosol size distributions in and out of cloud in sub-Arctic Finland showed that on average 30% of Aitken particles (defined as 25-95 nm) were activated to form droplets (Komppula et al., 2005). In Svalbard, measurements of cloud residuals (Karlsson et al., 2021) and cloud drop number concentrations (Koike et al., 2019) have been used to show the activation of particles smaller than 50 nm diameter during periods of high supersaturation, such as when updraft speeds are high or when accumulation mode concentrations are low. Results from parcel models and large-eddy simulation (LES) models are in agreement with the observations that Aitken particles can act as CCN (Pöhlker et al., 2021; Bulatovic et al., 2021), though Bulatovic et al. (2021) find that two sets of aerosol conditions under which Aitken activation is favourable in their model have real-world occurrence probabilities of 5 and 17%, raising questions about how widespread the phenomenon could be. Activation of Aitken particles means that particles formed by NPF and subsequent growth may only need to grow up 20-50 nm diameter to act as CCN in Arctic clouds, making it more plausible that such particles could survive long enough in the atmosphere to be important for cloud formation.

A further complicating factor that affects our understanding of summertime Arctic aerosol processes is the decoupling of the surface, where most aerosol measurements have been made in the high Arctic, and the cloud layer, where the aerosols act as CCN. Such decoupling is caused by the thermodynamic structure of the high Arctic summer boundary layer. Regimes of turbulence at the surface and in the cloud mixed layer have previously been shown to occur in configurations where the two regimes do not interact, inhibiting transport of moisture or particles vertically between the different layers (Shupe et al., 2013; Sotiropoulou et al., 2014; Brooks et al.). Decoupling of the surface in this way has implications for aerosol-cloud interactions because it implies that the aerosol sources, concentrations and size distributions at the surface may only be relevant for the cloud layer during sporadic events of mixing. This is very different from dynamics typical of lower latitudes, where strong convection can promote mixing of heat, moisture an aerosols from the surface up to higher altitudes. Models have struggled to capture decoupling in the Arctic, showing a tendency to become coupled too often (Birch et al., 2012; Sotiropoulou et al., 2016).

Recent observations from the high Arctic provide a new opportunity to explore the questions surrounding the source of summertime aerosol. The Arctic Ocean 2018 (AO2018) expedition took place in August and September 2018 between Svalbard and the North Pole (Vüllers et al., 2020; Leck et al., 2020). The observed properties and behaviour of the aerosol challenge our current understanding of aerosol sources and sinks. Firstly, mass spectrometry measurements clearly show that iodic acid is the main driver of NPF events, which occur primarily during sea ice freeze-up (Baccarini et al., 2020a). This process was not included in previous large-scale modelling studies. Secondly, the time series of particle concentrations and size distributions, spanning several weeks over the end of the sea ice melt period and transition to freeze-up, provide an opportunity to understand how sources and sinks are related to the melting/freezing cycle and changes in photochemistry. In particular, the observations show a distinct transition in aerosol behaviour in late summer, when the iodic acid NPF events begin to take place. Here we aim to interpret this change in behaviour in terms of changes in the dominant aerosol sources.

In this study we used measurements from AO2018 to evaluate the Arctic aerosol budget in the global climate model UKESM1. We compared the accuracy of simulations with NPF in the boundary layer to that of simulations with NPF in the free troposphere. We also introduced an iodic acid NPF scheme to the model to investigate iodic acid as an Arctic aerosol source compared to other components of the Arctic aerosol budget. The observational datasets are introduced in Sect. 2. The model is described in Sect. 3, including the different NPF schemes and our approach to the inclusion of iodic acid. Results are presented in Sect. 4 and discussed in Sect. 5.

## 2  Observations

We use data from three campaigns in different years: Arctic Ocean 2018 (AO2018), The Arctic Summer Cloud Ocean Study (ASCOS) in 2008, and the Atmospheric Tomography Mission (ATom) in 2016. We only use model output from a simulation of the year 2018 because the Arctic Ocean 2018 dataset is the main focus of this study. The interannual variability in aerosol concentration may limit how representative observations from 2008 (ASCOS) or 2016 (ATom 1) can be to assess model output from 2018. However, as we will show in later sections, the difference in particle concentration from different simulations in this

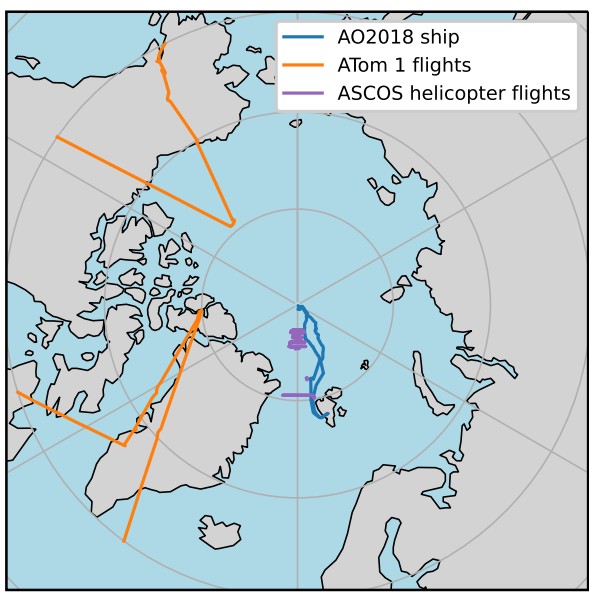

**Figure 1.** Map showing location of where observations where measured. Co-ordinates on map are for *Oden* during AO2018 (blue line), aircraft during leg 1 of ATom (orange line) and helicopter during ASCOS (purple line). Note only ATom 1 co-ordinates with a latitude greater than 60°N are shown here since we discarded data from further south in this study.

study can be several orders of magnitude and therefore is likely to exceed the range of concentrations that would be measured in different years. Simulated values of $N_{50}$ (number concentration of particles larger than 50 nm diameter) over a 30 period suggest that the interannual variability of $N_{50}$ in the Arctic does not account for such large differences in our simulated particle concentrations (Carslaw and Pringle, 2022).

## 2.1 Arctic Ocean 2018

The Arctic Ocean 2018 (AO2018) expedition took place in August and September 2018 aboard the Swedish icebreaker *Oden*. The ship travelled from Svalbard to the North Pole and then drifted, moored to an ice floe, for 4 weeks before travelling back to Svalbard (Fig. 1). Further details and meteorological conditions of the campaign are presented in Vüllers et al. (2020) and Leck et al. (2020). Here, we compare model output to aerosol and gas-phase measurements, which were measured during AO2018 as part of the Microbiology-Ocean-Cloud Coupling in the High Arctic (MOCCHA) campaign.

We use two aerosol datasets from the AO2018 campaign, involving three different instruments (Baccarini et al., 2020a). A differential mobility particle sizer (DMPS) measured particles in the size range 10–959 nm (Karlsson and Zieger, 2020). We integrate this size distribution over the ranges 15–100 nm and 100–500 nm. Total particle concentration for diameters greater than 2.5 nm was measured by an ultrafine condensation particle counter (UCPC). The UCPC and integrated DMPS data were used to calculate the concentration of all particles in the size range 2.5–15 nm (Baccarini and Schmale, 2020). This 2.5–15

nm time series was also supplemented by a particle size magnifier during periods when the UCPC was not in operation. All data has been selected for "clean" periods, i.e. excluding periods where the ship's exhaust might influence the measurements. In addition to the aerosol data, iodic acid concentrations were measured using a nitrate chemical ionization mass spectrometer (Baccarini et al., 2020b).

The measurements of nucleation mode particle concentration at the surface during AO2018 show marked difference in aerosol behaviour linked to the onset of local sea ice freeze-up. During the freeze period, peaks in the nucleation mode concentration occur during NPF events, lasting on the order of hours (see section 4.2). Thus the nucleation mode concentration is on average higher in the freeze period than the melt period, and also fluctuates more.

## 2.2 Arctic Summer Cloud Ocean Study 2008

The Arctic Summer Cloud Ocean Study (ASCOS) campaign also took place on the icebreaker *Oden*, at roughly the same time of year as AO2018 but a decade earlier (2008). The ASCOS drift period was from $12^{th}$ August until $2^{nd}$ September 2008 and took place close to 87°N. A full description of the campaign is given in Tjernström et al. (2014).

A helicopter was used during ASCOS to take measurements above the surface, with two condensation particle counters (CPCs) and one optical particle counter (OPC) used to measure aerosol concentrations (Leck et al., 2022). The CPCs and OPC detected particles larger than 3 nm, 14 nm, and 300 nm respectively, giving an overall aerosol size distribution in the ranges 3–14, 14–300 and >300 nm. These do not exactly match the aerosol size ranges available from the AO2018 data, but still give us valuable information about the aerosol size distribution from this campaign. The aerosol measurements are presented in Kupiszewski et al. (2013).

## 2.3 Atmospheric Tomography Mission

The Atmospheric Tomography Mission (ATom) was a multi-year flight campaign that used the NASA DC-8 aircraft to study the effects of air pollution on the chemistry of the atmosphere. There were four legs of ATom, each carried out in a different season in different years. Here, we use measurements taken during the first leg, which took place in summer 2016. Measurements were taken over a wide range of latitudes and altitudes. We only use measurements taken north of 60°N.

Aerosol size distributions were measured during ATom using a nucleation-mode aerosol size spectrometer (NMASS), an ultra-high-sensitivity aerosol size spectrometer (UHSAS) and a laser aerosol spectrometer (LAS). The full set of ATom aerosol measurements are presented in Brock et al. (2019). Aerosol size distributions are available for particles between approximately 3 nm and 3.5 μm diameters.

Note that the AO2018, ASCOS and ATom campaigns use instruments with different ways of sizing particles. The DMPS measures the electrical mobility diameter of particles. The CPCs and NMASS measure particles based on the critical diameter for droplet nucleation at the instrument's operating supersaturation. The OPC, UHSAS and LAS all rely on optical methods to measure particles, meaning they measure the optical equivalent diameter of particles. No attempt has been made to convert from optical equivalent diameter to mobility diameter. Such a conversion would require information about the refractive indices of the particles measured by the optical instruments, which is not available.

## 3 Model description

We used the UK Earth System Model version 1 (UKESM1, Mulcahy et al., 2020; Sellar et al., 2019) in its atmosphere-only configuration, which uses output from a fully-coupled run of UKESM1 to prescribe sea surface temperatures (SSTs) and some biogenic emissions, such as DMS, from the ocean. The dynamical core of the model is the UK Met Office Unified Model (UM) in global atmosphere configuration version 7.1 (GA7.1, Walters et al., 2019) with a horizontal resolution of 1.875° longitude by 1.25° latitude and 85 vertical levels. The vertical resolution is approximately 50 m at the surface, 150 m at 1 km altitude and on the order of km at the highest model level. We ran the model in its "nudged" configuration, which means horizontal winds and potential temperature are relaxed to ERA-Interim values on a 6-hourly time scale above approximately 1 km. The model simulates gas and aerosol chemistry using the UK chemistry and aerosols model (UKCA, Morgenstern et al., 2009; O'Connor et al., 2014). Aerosol microphysics is simulated by the Global Model of Aerosol Processes (GLOMAP-Mode, Mann et al., 2010) using 5 log-normal aerosol size modes (4 soluble and 1 insoluble). Mode sizes and geometric widths are given in Table 1. The model uses 2-moment aerosol microphysics, meaning that the number and mass in each mode are prognostic variables. There are four aerosol species: sulphate, organic carbon, black carbon and sea salt. Processes handled by GLOMAP include primary emissions, coagulation within and between modes, condensational growth and ageing, new particle formation, dry deposition, wet deposition within and below clouds, and aqueous sulphate production in cloud droplets.

Emissions of $SO_2$, black carbon (BC) and organic carbon (OC) are for the year 2014 for all emissions sectors except biomass burning, which are from a climatology of the years 1995-2004. We use a climatology for biomass burning to minimise any potential bias from boreal forest fire emissions not being from the same year as the simulation time. The emissions datasets used for aerosols and precursor vapours are Hoesly et al. (2018) ($SO_2$, anthropogenic OC and BC), Marle et al. (2017) (biomass burning OC and BC) and Sindelarova et al. (2014) (monoterpenes). Sea salt emissions are calculated using the Gong (2003) parameterisation. Primary marine organic emissions are then calculated using the sea spray flux, 10 m wind speed and chlorophyll-a concentration, using the Gantt et al. (2015) parameterisation. Note that because we use an atmosphere-only configuration in this study, the chlorophyll-a concentration is taken from an ancillary file, produced using model output from the fully-coupled model. Marine emissions are scaled by grid box open water fraction for sea ice regions.

### 3.1 New particle formation schemes

UKESM includes binary homogeneous nucleation of water and sulphuric acid ($H_2SO_4$) using the parameterisation of Vehkamäki et al. (2002). The Vehkamäki et al. (2002) scheme creates particles mostly in the cold free and upper troposphere. Here, we run simulations with NPF schemes that also create particles in the boundary layer (BL). We do this so that we can compare NPF driven by iodic acid, observed during AO2018, to other nucleation mechanisms known to be important in the Arctic and extra-Arctic. We simulated nucleation of $H_2SO_4$ in the BL by cluster activation as described by Kulmala et al. (2006). We simulated organically mediated $H_2SO_4$ nucleation using the parameterisation of Metzger et al. (2010). We refer to simulations by the precursor vapour used to drive new particle formation in each simulation, i.e. SA for the use of sulphuric acid in the Kulmala et al. (2006) scheme, SOA for secondary organic vapours in Metzger et al. (2010), or IA for iodic acid.

**Table 1.** Description of aerosol modes and parameterisations in the model. Y/N=yes/no i.e. if Y, this species is allowed to exist in this mode. J96=Jacobson et al. (1996).

| | Nucleation | Aitken | Accumulation | Coarse | Insoluble | Water uptake |
|---|---|---|---|---|---|---|
| **Size (nm)** | 1–10 | 10–100 | 100–500 | 500–1000 | 10–100 | |
| **Width** | 1.59 | 1.59 | 1.4 | 2 | 1.59 | |
| **Sulphate** | Y | Y | Y | Y | Y | As for $SO_4$ in J96 |
| **Organic carbon** | Y | Y | Y | Y | Y | 65% that of sulphate |
| **Black carbon** | N | Y | Y | Y | Y | None |
| **Sea salt** | N | N | Y | Y | N | As for Cl in J96 |
| **Hygroscopic growth** | Following ZSR theory as described in Mann et al. (2010). | | | | | |
| **Aerosol activation scheme** | Abdul-Razzak and Ghan (2000) | | | | | |

The formation rates of 1.5 nm clusters, $J_*$, used by M10 and K06 are given in cm$^{-3}$ s$^{-1}$ by

$$J_{*,K06} = k_{K06} C_{SA} \tag{1}$$

$$J_{*,M10} = k_{M10} C_{SA} C_{SOA} \tag{2}$$

where $k_{K06} = 10^{-6}$ s$^{-1}$, $k_{M10} = 10^{-13}$ cm$^3$ s$^{-1}$, and $C_x$ are the concentrations of the precursor vapours in cm$^{-3}$. The 3 nm particle formation rate is calculated from the 1.5 nm cluster formation rates using the method from Kerminen and Kulmala (2002), which accounts for growth of clusters and loss to existing particles.

It should be noted that these two BL NPF schemes use empirical parameterisations that have not been developed with or tested against data from the Arctic. Moreover, the parameterisations only consider the influence of $H_2SO_4$ and secondary organic vapours. MSA and ammonia are not included in UKESM and thus are not modelled as NPF precursors in this study. These omissions are potential sources of bias given the importance of MSA and ammonia in the Arctic (Willis et al., 2018; Beck et al., 2020).

As we show below, entrainment of nucleation mode aerosol from the FT is a key surface aerosol source in the central Arctic in UKESM. To investigate the role of FT entrainment as a source of surface particles in the high Arctic, we ran an additional sensitivity test with an imposed NPF rate in the low FT (simulation SOA_PRSC). A particle formation rate of $10^{-2}$ cm$^{-3}$ s$^{-1}$ was used for altitudes above the top of the BL and below 7.5 km, and for latitudes north of 80°N. Since this rate is higher than what the Metzger et al. (2010) scheme typically produces, it allows us to investigate the sensitivity of the surface concentrations to a strong free troposphere source.

## 3.2 Iodic acid

An empirical model for the steady-state concentration of iodic acid has been produced from observations taken in the high Arctic (Baccarini et al., 2020a). The concentration in cm$^{-3}$, $C_{IA}$, is given by

$$C_{IA} = \frac{E}{v_{dep} + h \cdot CS} \tag{3}$$

where $E$ is the emission rate of iodine atoms in cm$^{-2}$ s$^{-1}$, $v_{dep}$ is the dry deposition velocity of iodic acid in cm s$^{-1}$, $h$ is the surface mixed layer height in cm and $CS$ is the condensation sink due to all aerosols (including the nucleation mode formed through this process), given in s$^{-1}$. $E$ can be considered a net emission rate, which also accounts for the conversion of iodine into HIO$_3$ (the rate is assumed to be constant). This paremeterisation does not include the HIO$_3$ chemical formation mechanism described in Finkenzeller et al. (2022). Still, it can reproduce the observed iodic acid concentration, as shown in Figure 2 and already reported in Baccarini et al. (2020a). Hence, it serves well the scope of this work. It was observed during AO2018 that fog and cloud droplets acted as a strong sink of the iodic acid, however we do not account for that here since the parameterisation of clouds in the coarse resolution model gridboxes is unlikely to be representative of conditions at the ship.

We used Eq. (3) to diagnose the steady-state concentration of iodic acid in model gridboxes at each timestep. The iodic acid was then used as a precursor to drive NPF, a process which also depletes the iodic acid gas concentration. The dry deposition velocity of iodic acid was calculated by the model assuming a diffusion constant equal to that of H$_2$SO$_4$. The condensation sink due to existing aerosols was calculated during model runs using the aerosol size distribution produced by the model. We use the modelled dry aerosol diameter in the condensation sink calculation, meaning that the effects of water uptake by aerosol are not included. For the surface mixed layer height, we take the BL height calculated by the UM. Finally, we use a value of $5.21 \times 10^6$ cm$^{-2}$ s$^{-1}$ for $E$, which is approximately equal to the median of the distribution of $E$ measured by Baccarini et al. (2020a).

The concentration of iodic acid was observed to increase towards the end of summer, during the sea ice freeze-up (Baccarini et al., 2020a). It is thought that the freezing of sea water can trigger the emission of iodine from the surface. To incorporate this behaviour into UKESM, we calculate the concentration of iodic acid in gridboxes where sea ice fraction is non-zero and where the surface temperature is less than -5 °C. Although the observed freeze-up date is defined as the day when the 14-day running mean of surface temperature reaches -2 °C (the temperature at which sea water freezes), we found that -5 °C acted as a better threshold for ice freeze-up in the model. This is because there is a cold bias in the model such that when the observed 14-day running mean temperature reaches -2 °C, the modelled surface temperature is closer to -5 °C (see Fig. A1). The calculated concentration of iodic acid is then equally distributed from the surface the BL height. This is equivalent to assuming that the lifetime of iodic acid is long enough for the gas to be mixed throughout the BL.

It is important to note that a direct/causal mechanism linking the freeze-up to enhanced iodine emissions has not yet been identified. However, the results of our study would remain valid even if the two processes were not directly related. In fact, the surface temperature threshold used in the model is a good tracer for the summer to autumn transition, which has been associated with higher iodine concentration in the Arctic (Baccarini et al., 2020a; Sharma et al., 2019)

The particle formation rate at 3 nm from iodic acid is calculated using the method from Kerminen and Kulmala (2002) as for the $H_2SO_4$ activation and organically mediated schemes, using a kinetic rate of cluster formation from iodic acid. This is in line with recent results from cloud chamber experiments (He et al., 2021). The cluster formation rate in $cm^{-3}$ $s^{-1}$ is given by

$$J_{*,IA} = k_{IA}C_{IA}^2 \qquad (4)$$

where $k_{IA} = 10^{-13}$ $cm^3$ $s^{-1}$. The mass created from $HIO_3$-driven new particle formation is added to the sulphate model component.

## 3.3 Secondary organic vapours

Assumptions about the production of secondary organic aerosol material have a large effect on modelled Arctic aerosol. Secondary organic aerosol is created in the model by the oxidation of monoterpenes by ozone, the hydroxyl radical and the nitrate radical (Spracklen et al., 2006). These reactions produce secondary organic aerosol on the timescale of hours. This is a simplification of the process, since in reality there are many chemical species and reaction pathways involved in the production of secondary organics, which operate on a range of timescales and produce a range of volatilities. In particular, some organic aerosol precursor species have a longer lifetime than monoterpene and can therefore be transported further in the atmosphere before forming aerosol and condensing on existing aerosol. In the model, most monoterpene is oxidized close to the source region (e.g. boreal forests) and quickly condenses onto existing particles, therefore the concentration of organic aerosol precursor gases is very low in the central Arctic; as a consequence, NPF involving organic vapours is extremely weak. To investigate the effect of these assumptions on Arctic NPF, we ran sensitivity simulations where the oxidation rate of monoterpene was reduced by a factor of 100. The labels of these simulations use the suffix _OXID.

We alter the oxidation rate of monoterpene to promote transport of monoterpenes north, to account for missing species and reactions that create organic aerosol precursors. This approach allows us to test the effect of neglected organic species with oxidation rates different from monoterpenes.

## 3.4 Ageing of insoluble particles

The default assumptions in UKESM about particle ageing have a very substantial effect on Arctic aerosol, particularly where NPF is a major source of Aitken mode particles. Particle ageing in a model is the transfer of insoluble particles into the soluble particle modes after condensation of water-soluble material. In UKESM, sources of insoluble carbonaceous particles are biofuel and biomass burning emissions (mean diameter 150 nm), fossil fuel burning emissions (mean diameter 60 nm) and primary marine organic carbon emissions (mean diameter 160 nm). By default, the aged mass from the insoluble mode is moved into the soluble Aitken mode (Mulcahy et al., 2020). However, the mean diameter of the insoluble mode does not usually correspond to the size limits of the soluble Aitken mode, and this can lead to undesirable behaviour in the model. Thus, when mass from the insoluble mode is moved into the soluble Aitken mode, it will typically increase the mean diameter of the soluble mode beyond its upper limit (100 nm). When small particles enter the Aitken mode following growth of nucleation

**Table 2.** Description of simulations.

| Simulation | Description |
|---|---|
| **Main body** | |
| CONTROL | Model set-up based on UKESM1 atmosphere-only configuration. Binary $H_2SO_4$-water vapour NPF parameterised as per Vehkamäki et al. (2002). |
| SOA | Organically-mediated NPF (driven by secondary organic aerosol precusors) in the BL, using Metzger et al. (2010) |
| SOA_PRSC | Organically-mediated NPF in all model levels, using Metzger et al. (2010), and a prescribed NPF rate of $10^{-2}$ $cm^{-3}$ $s^{-1}$ between the BL top and 7.5 km and north of 80°N |
| IA | Additional BL NPF driven by IA |
| IA_SOA | Organically-mediated NPF in all model levels, using Metzger et al. (2010), with additional BL NPF driven by IA |
| SOA_85N | Additional NPF in the BL using Metzger et al. (2010) for gridboxes north of 85 °N only |
| **Appendices** | |
| SA | Additional BL $H_2SO_4$ NPF parameterised by Kulmala et al. (2006) |
| SOA_ALL_LEVELS | Organically-mediated NPF in all model levels, using Metzger et al. (2010) |
| XXX_OXID | Additional change to oxidation of monoterpenes as described in section 3.3 |

mode particles, they are artificially strongly depleted because "mode merging" (Mann et al., 2010) requires that their mass is averaged with the larger particles already existing in the Aitken mode. This combination of assumptions (the size of aged particles and mode merging) is adequate for reproducing size distributions in the mid-latitudes, where anthropogenic and fire emissions are a more dominant source. However, our early simulations showed that this method has a very substantial effect in the Arctic where NPF is occurring in air that has aged during long-range transport from low latitudes to the Arctic (see appendix C). In particular, we found that the particle size distribution was extremely insensitive to the NPF rate. We therefore altered the model such that aged carbonaceous particles are moved directly into the soluble accumulation mode (100 – 500 nm), as is appropriate for their diameter. Our CONTROL simulation uses this altered ageing scheme. Simulations using the model's default ageing scheme are presented in appendix C.

A description of all model simulations is given in Table 2.

## 4   Results

We organise the results as follows. First we examine the behaviour of our empirical iodic acid scheme by comparing modelled surface iodic acid concentrations to observations in Sect. 4.1. Then in Sect. 4.2 we consider the effect of BL NPF in the model on the surface aerosol concentration during AO2018. Since we show that BL NPF is not taking place locally during the melt period of AO2018, we consider FT NPF as a source of surface particles in Sect. 4.3, using surface observations from AO2018

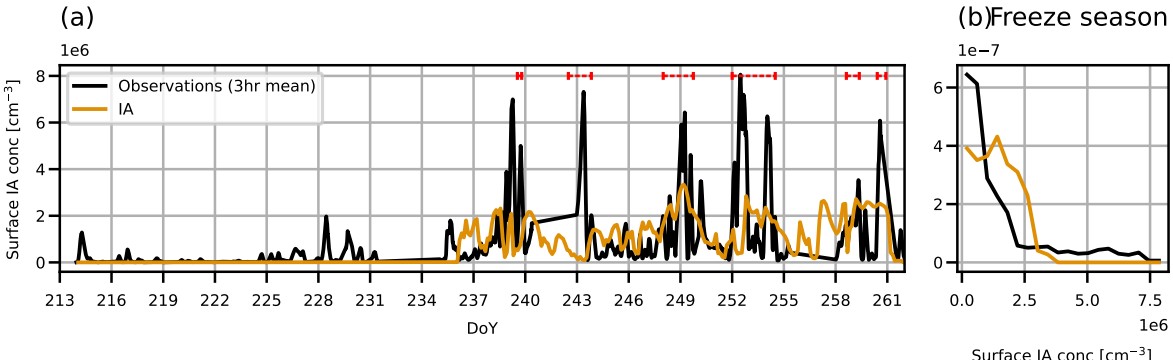

**Figure 2.** Times series and PDF of surface iodic acid concentration during AO2018 from observations and model output. Model output is from simulation IA (orange lines). Observations are shown as 3-hourly mean (black lines). Red dashed lines in (a) show periods of observed NPF events. PDF of concentration is for freeze season only, i.e. after 27th August 2018 (day 239).

and aerosol profiles from ASCOS in 2008 and ATom in 2016. Finally, we consider a combination of BL and FT sources in Sect. 4.4.

When comparing model output to observations of surface aerosol concentration, we use the overlap index defined in Pastore and Calcagnì (2019) to quantify the similarity between the distribution in aerosol concentration from observations and different model simulations. For two probability density functions $A(x)$ and $B(x)$, the overlap index $\eta(A, B)$ is defined as

$$\eta(A, B) = \int \min[A(x), B(x)]\, dx \tag{5}$$

which can be thought of as integrating the area where the distributions overlap. If the two distributions overlap completely, the entire area under the distribution will be integrated so the index will be 1, whereas if they do not overlap at all, no area will be integrated and the index will be 0. Overlap indices closer to 1 therefore indicate simulations with good model-observation agreement in terms of the magnitude and variability of the two time series even if individual peaks and troughs do not match temporally. In our case, we apply the overlap index to discrete distributions of binned aerosol concentration, so the integral becomes a sum.

### 4.1 Iodic acid concentration

Figure 2 shows a time series and PDF of surface $HIO_3$ concentration from observations and model output. To colocate the model output with the ship, we take the model gridbox nearest the ship's position. In the observations, the surface $HIO_3$ concentration is lower in the melt period (before day 239) than in the freeze period (after day 239). The surface concentration in the freeze period has a baseline of approximately $10^6$ cm$^{-3}$ and peak values 5–6 times higher lasting on the order of hours, whereas in the melt period the concentration reaches $10^6$ cm$^{-3}$ only for brief periods, such as on day 228. The periods of peak surface concentration in the freeze period correspond to observed NPF events.

**Table 3.** Overlap indices calculated for the PDFs of nucleation, Aitken and accumulation aerosol concentrations from observations and each simulation, measured at the surface. PDFs are separated into the melt and freeze periods before overlap indices are calculated. Bold text indicates the greatest overlap index in each mode for the melt and freeze periods.

| Simulation | Nucleation | | Aitken | | Accumulation | |
|---|---|---|---|---|---|---|
| | Melt | Freeze | Melt | Freeze | Melt | Freeze |
| CONTROL | 0.01 | 0.00 | 0.31 | 0.12 | 0.57 | 0.64 |
| SOA | **0.73** | 0.05 | 0.73 | 0.34 | 0.50 | 0.55 |
| SOA_85N | 0.01 | 0.00 | 0.33 | 0.13 | 0.56 | **0.67** |
| SOA_PRSC | 0.35 | **0.52** | **0.75** | 0.37 | 0.51 | 0.55 |
| IA | 0.25 | 0.47 | 0.35 | 0.70 | **0.58** | 0.58 |
| IA_SOA | 0.63 | 0.51 | 0.69 | **0.77** | 0.55 | 0.56 |

In the model, $HIO_3$ is only emitted when the surface temperature is -5 °C or less. The modelled temperature reaches this threshold on approximately day 236 at the ship's position, such that the model starts to emit $HIO_3$ at a similar time to the observed freeze-up onset (day 239), when $HIO_3$ concentrations were observed to increase at the ship. After $HIO_3$ starts to be emitted in the model, the empirical scheme we use for $HIO_3$ production consistently calculates surface $HIO_3$ concentration to be the same order of magnitude as the observations. The PDF in Fig. 2(b) is for the freeze season only, i.e. day 239 onward. The observed distribution of surface $HIO_3$ concentration is broadly captured, though the model does not reproduce the highest or lowest observed concentrations. This is likely to be because the model does not resolve the spatially heterogeneous sea ice state (which controls the emissions of iodic acid) nor the variability in clouds and fog, which control iodic acid scavenging as well as influencing the condensation sink due to existing aerosols.

## 4.2 Effect of NPF in the boundary layer

### 4.2.1 Time series of particle concentrations

Figure 3 shows time series and PDFs of surface aerosol concentration in three particle size ranges from the AO2018 observations and for simulations CONTROL, SOA and IA. Observations are from the ship (approximately 15 m above the ground) and model output is for the first model level, between approximately 0 and 37 m.

A time series of the measured nucleation mode (2.5–15 nm diameter) particle concentration during AO2018 is shown in Fig. 3 (a). Periods where iodic acid NPF events were observed are marked with red dashed lines. There is a marked difference in the behaviour of the nucleation mode concentration before and after the onset of sea ice freeze-up (27th August 2018, day 239). In the melt period (up to day 239), the nucleation particle concentration rarely exceeds 100 $cm^{-3}$ and is usually between 1–10 $cm^{-3}$. The NPF events, which occurred after the freeze-up began in the vicinity of the ship on day 239, are associated with peaks in the nucleation mode particle concentration lasting a few hours, causing fluctuations in particle concentration between approximately 10 and $10^4$ $cm^{-3}$, consistent with a strong local source.

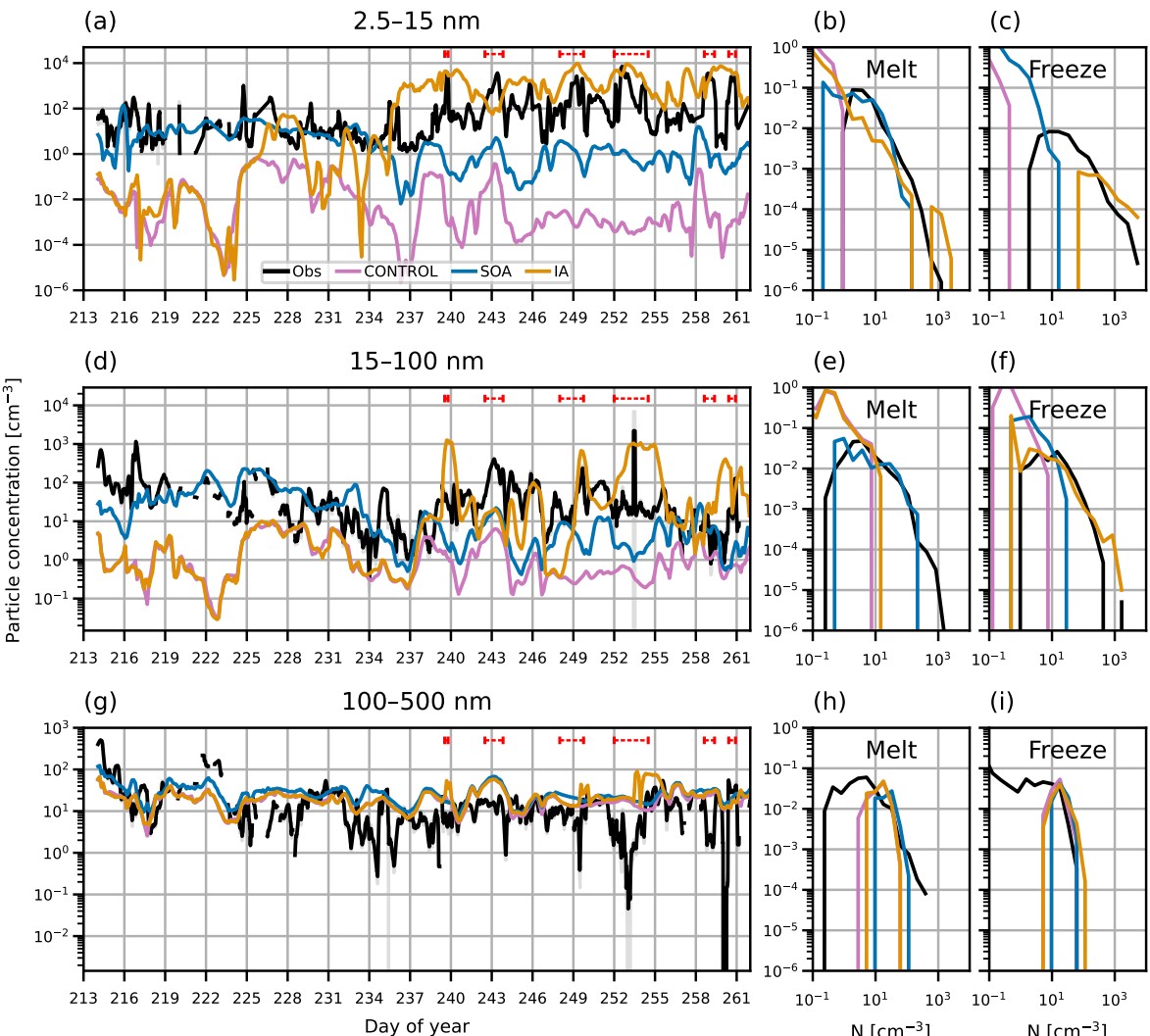

**Figure 3.** Times series and PDFs of surface aerosol concentration during AO2018 from observations and model simulations with various boundary layer NPF mechanisms. Model output is from simulations CONTROL (pink lines), SOA (blue lines) and IA (orange lines). Observations are shown as 3-hourly mean (black lines) and standard deviation (grey shading). Aerosol concentrations are shown for particles with diameter (a-c) 2.5-15 nm, (d-f) 15-100 nm and (g-i) 100-500 nm. Red dashed lines in (a, d, g) show observed NPF events. PDFs are separated by observed sea ice freeze-up date, 27th August 2018 (day 239).

The modelled nucleation mode particle concentration is underestimated in simulations CONTROL and IA in the melt period, while the inclusion of $HIO_3$ NPF means that IA performs better in the freeze period than during the melt. The nucleation mode concentration in CONTROL varies from roughly $10^{-6}$ to $10^{-1}$ cm$^{-3}$ and is usually at least an order of magnitude lower than observed throughout the whole period. In contrast, IA produces nucleation mode concentrations of roughly the correct

order of magnitude during the freeze period (day 239 onwards), with a distinct change in concentration and behaviour around day 235. However, the lack of $HIO_3$ emissions during the melt period means that the model continues to underestimate the

concentration by several orders of magnitude. The model underprediction of nucleation mode particle concentration in the melt period is an indication that $HIO_3$ NPF is not the only part of the regional aerosol budget that needs consideration to produce an accurate simulation. Simulation SOA uses a BL NPF scheme and has higher nucleation mode concentrations than CONTROL as a result. In the melt period, SOA consistently simulates nucleation mode concentrations of the same order of magnitude as the observations. In the freeze period, SOA still underestimates the nucleation mode concentration despite simulating higher

concentrations than CONTROL.

Overlap indices for the PDFs of observed surface aerosol concentration and modelled concentration from all simulations are given in Table 3. The underestimation of nucleation mode concentration by simulation CONTROL is highlighted by the fact that its overlap indices are close to 0 in both the melt and the freeze periods. The SOA PDFs of nucleation mode concentration for the melt period match the observations better than CONTROL, producing an overlap index of 0.73. However, in the freeze

period, the overlap index is low at 0.05. IA performs moderately better than CONTROL in both periods, giving overlap indices of 0.25 in the melt period and 0.47 in the freeze period.

Observed Aitken mode concentrations (diameter 15–100 nm) lie between about 1 and 1000 $cm^{-3}$, with a a mean of 60 $cm^{-3}$ over the whole period. In contrast, concentrations in the CONTROL simulation are around 1 $cm^{-3}$, frequently fall below 0.1 $cm^{-3}$, and never exceed 10 $cm^{-3}$. The simulation with $HIO_3$ again shows a sharp increase in particle concentration around

395   day 235 when $HIO_3$ starts being emitted during the freeze period. Model Aitken mode concentrations then vary between being comparable to the observations and being 1–2 orders of magnitude too low. As in the nucleation mode, the Aitken mode concentrations from SOA are higher than CONTROL, on the same order of magnitude as the observations in the melt period but underestimating observations in the freeze period.

The observed accumulation mode concentrations (diameter 100–500 nm) are typically around 10–100 $cm^{-3}$, with brief

periods of less than a day where they fall to 1 $cm^{-3}$ or lower, which is characteristic of the central Arctic (Bigg et al., 1996; Bigg and Leck, 2001; Mauritsen et al., 2011; Leck and Svensson, 2015). All simulations capture the observations well, except for the periods of extremely low concentration. The good agreement shows that NPF is not an important source of accumulation mode aerosol. As shown in e.g. Stevens et al. (2018) and Loewe et al. (2017), the periods of very low concentration are associated with efficient scavenging in drizzle on smaller spatial scales than represented in a global model. Nevertheless, the generally

good model-observation agreement means that the aerosol surface area, and hence condensation sink for nucleating vapours, is reasonable in the model, and therefore not a cause of the biases in nucleation and Aitken mode particle concentrations.

### 4.2.2   Aerosol vertical profiles

In Sect. 4.2.1, we showed that the use of organically-mediated BL NPF in the model (simulation SOA) instead of only the default NPF scheme (simulation CONTROL) increased nucleation and Aitken particle concentrations at the surface during the

melt period of AO2018. In this section, we examine the effect of this scheme on particle concentrations aloft, and show that switching on BL NPF increases particle concentrations in the Arctic FT, perhaps counter-intuitively.

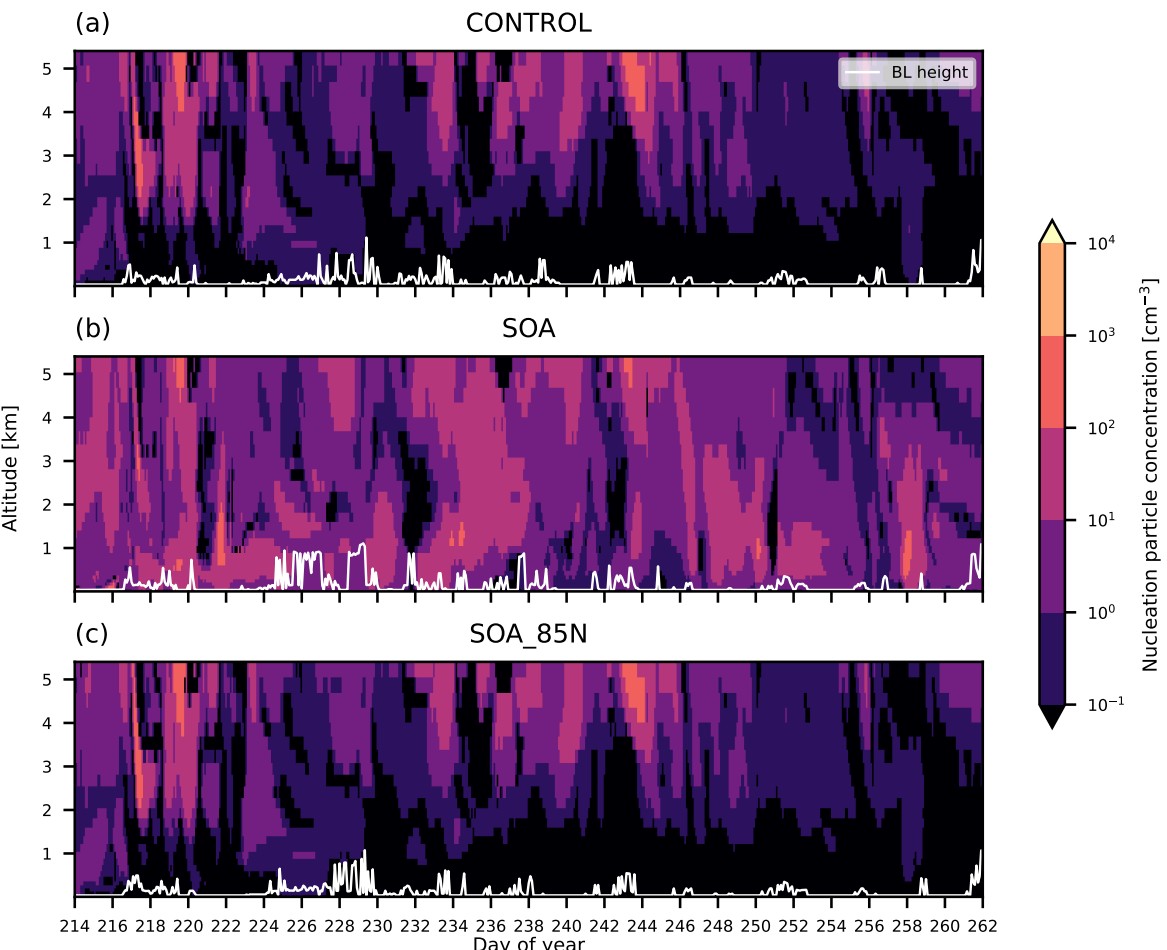

**Figure 4.** Nucleation mode aerosol profiles simulated for AO2018 campaign period. Model output is from simulations (a) CONTROL, (b) SOA and (c) SOA_85N. Model output was colocated with the position of the *Oden*. White lines show BL height.

Figure 4 shows simulated nucleation mode aerosol vertical profiles for the AO2018 campaign period. As discussed in Sect. 4.2.1, the nucleation mode concentrations at the surface in CONTROL are typically less than 0.1 cm$^{-3}$ and never above 1 cm$^{-3}$. We showed that this is an underestimation of the measurements taken at the ship. Figure 4 shows that despite the low concentrations at the surface, the model produces nucleation mode concentrations of up to 1000 cm$^{-3}$ in the low FT, for example on day 244 above approximately 4 km. These higher concentrations aloft suggest that NPF is being simulated in the FT, but that those particles do not reach the surface.

The vertical profile for simulation SOA shows that the nucleation mode concentration is higher than CONTROL above the BL as well as at the surface, even though this simulation is using the same NPF mechanism as CONTROL in the FT. We previously showed that SOA had nucleation mode concentrations at the surface that were 1–3 orders of magnitude higher than

that of CONTROL. In the FT, the increase is up to an order of magnitude. The white lines in Fig. 4 show the top of the BL in the model, which is used by the NPF schemes to separate the BL from the FT. The increased nucleation mode concentrations in SOA above this BL height suggest that particles are being created in the BL at lower latitudes and then transported north in the FT. To examine this effect further, we ran a simulation where the organically-mediated NPF scheme was used in the BL only for latitudes north of 85°(as opposed to globally, as it is in SOA). The nucleation mode vertical profiles for this simulation, SOA_85N, are also shown in Fig. 4 and show lower concentrations than SOA in both the BL and the FT. In fact, the nucleation mode concentration in SOA_85N is mostly the same as that of CONTROL. The output from SOA_85N therefore shows that the higher concentrations in SOA are from latitudes south of 85°. The concentration of organic vapours north of 85°N is much lower than at lower latitudes (see the steep latitudinal gradient in Fig. E4), which potentially accounts for the low particle concentrations in SOA_85N. Note that the smaller spatial extent of the NPF scheme could lead to less depletion of precursors than when the scheme is used globally, thereby increasing the NPF rate in SOA_85N relative to SOA. However, this non-linear behaviour would have the opposite effect to what we describe here so we can disregard it.

## 4.3 Effect of free-tropospheric NPF

In Sect. 4.2 we showed that the model default NPF scheme is insufficient to produce an accurate simulation of the aerosol concentrations measured at the surface during AO2018. The inclusion of iodic acid NPF improves the concentration in the nucleation and Aitken modes in the freeze period. Nevertheless, substantial model underestimations in particle concentration remain during the melt period, indicating some other missing source. A simulation with BL NPF driven by secondary organic vapours (simulation SOA) produced more accurate surface concentrations in the melt period of AO2018 compared to observations. However, as we showed in Sect. 4.2.2 using output from simulation SOA_85N, the SOA simulation does not produce more NPF in the BL in the high Arctic. Rather, small particles produced at lower latitudes are transported north, increasing nucleation and Aitken concentrations aloft as well as at the surface. This behaviour in the model raises the question that the FT could be a source of particles to the surface. Such a source has been considered before, for example by Korhonen et al. (2008a), who showed that entrainment of secondary particles from the FT is an important source of CCN over the Southern Ocean in GLOMAP, and Igel et al. (2017) who used a high resolution LES model to show that particles can be transported from the free troposphere to the surface under conditions typical of the high Arctic. Also of relevance to entrainment processes in Arctic clouds, Solomon et al. (2011) used an LES model to show that a humidity inversion and entrainment of water vapour at cloud top helps to maintain the cloud by supplying moisture.

In this section we explore the role of the FT as a source of particles at the surface in both periods. We ran a simulation with the Metzger et al. (2010) organically-mediated NPF scheme switched on at all model levels and with a fixed NPF rate for model levels between the top of the BL and 7.5 km (simulation SOA_PRSC). The use of an idealised, constant NPF rate above the top of the BL in SOA_PRSC tests the sensitivity of the surface aerosol concentration to a source of aerosols from the FT. We use the output from SOA_PRSC to examine whether particles from the FT are being entrained into the BL in the model.

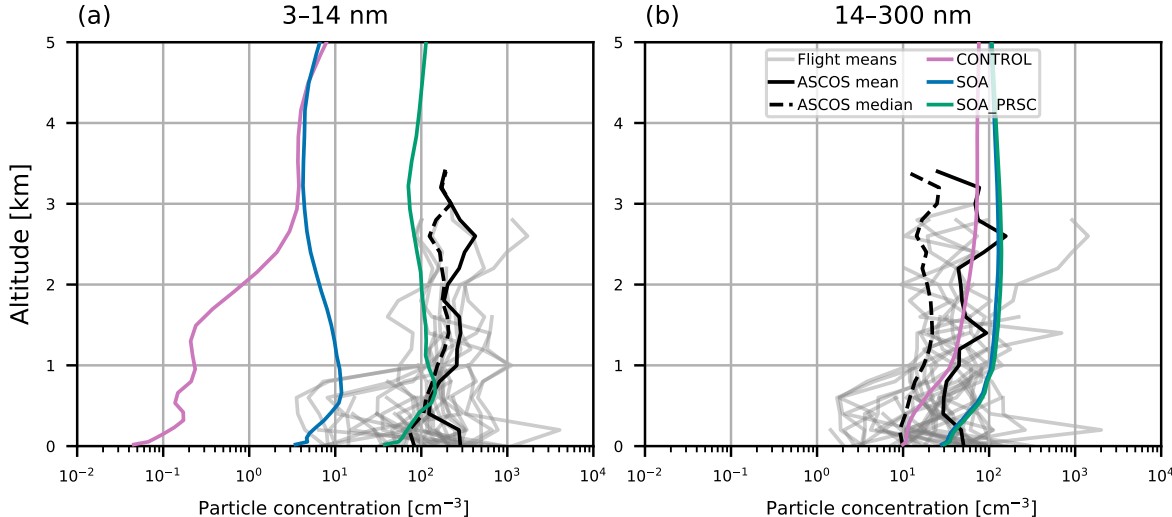

**Figure 5.** Aerosol vertical profiles from model output and ASCOS campaign observations from 2008. Model output is from colocated 2018 monthly mean values from simulations CONTROL (pink), SOA (blue lines) and SOA_PRSC (green lines). Observed values are given as the mean profile from each ASCOS flight (grey lines), overall mean (solid black) and overall median (dashed black). Profiles are for particles with size (a) 3-14 nm, measured during ASCOS using a UCPC and (b) 14-300 nm, measured using a CPC (particles greater than 14 nm) and a CLASP instrument (particles greater than 300 nm).

### 4.3.1 Aerosol vertical profiles

Figure 5 compares simulated aerosol vertical profiles against observations from ASCOS (see Sect. 2.2). Model particle concen-
trations in these size ranges were calculated using August 2018 monthly mean data, colocated with the ASCOS flights which
took place in August 2008. Concentrations of 3–14 nm diameter particles were consistently in the range $10^2$–$10^3$ cm$^{-3}$ in the
FT, with lower values recorded below 1 km. The CONTROL simulation fails to capture these concentrations, underestimating
the ASCOS mean concentration throughout the profile by 2 orders of magnitude in the FT and 3 orders of magnitude at the
surface. Inclusion of organic BL NPF (SOA) substantially increases particle concentrations, but they remain at least 1 order
of magnitude lower than observed. The model captures the 14–300 nm concentration better than the smaller particles. This is
consistent with the results from the AO2018 comparisons, where the accumulation mode was captured much better than the
nucleation or Aitken modes. All simulations shown here have 14–300 nm diameter particle concentrations of the same order
as those measured in the ASCOS flights.

Aerosol vertical profiles measured in 2016 during the ATom campaign are shown in Fig. 6 with model output for the year
2018. Consistent with the ASCOS and AO2018 datasets, the CONTROL simulation predicts 5–10 nm diameter particle con-
centrations up to 3 orders of magnitude lower than the ATom measurements in the lowest 5 km of the atmosphere, with the
largest model-observations discrepancies at the surface. In contrast to ASCOS, the SOA simulations captures the nucleation

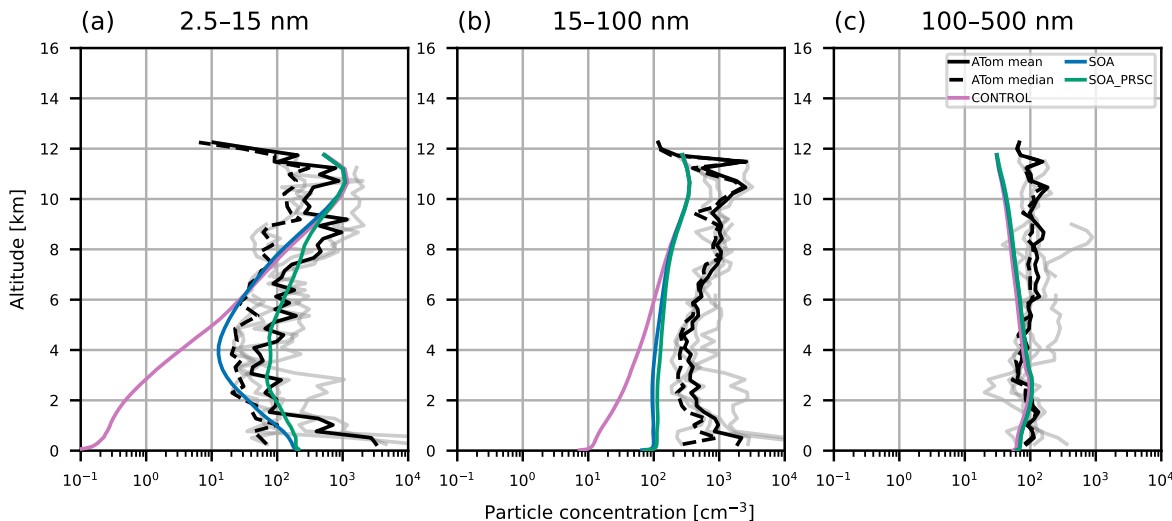

**Figure 6.** Aerosol vertical profiles from model output and ATom campaign observations from 2016. Model output is from colocated 2018 monthly mean values from simulations CONTROL (pink), SOA (blue lines) and SOA_PRSC (green lines). ATom observations are taken from leg 1 of the campaign and restricted to measurements that were taken north of 60°N. Observations correspond to mean profiles from different days (grey lines), the overall mean (black solid lines) and overall median (black dashed lines). Profiles are for particles with size (a) 5–10 nm, (b) 10–100 nm and (c) 100–500 nm. Observations were recorded at standard temperature and pressure, model output has been adjusted to account for this.

mode profile better, producing concentrations that are within the range of the ATom measurements. The Aitken mode is under-estimated by all simulations, though simulation SOA is closer to the observations than CONTROL in the lowest part of the FT.

All simulations capture the accumulation mode well at the surface but underestimate particle number aloft by roughly 50%.

SOA underpredicts the nucleation mode aerosol concentrations measured during the ASCOS and ATom campaigns. How-ever, SOA simulates higher nucleation mode concentrations than CONTROL. As shown in section 4.2.2, the higher concen-trations in the central Arctic in SOA occur as a result of transport of particles from further south, and concentrations are also increased in the FT by this transport. We therefore tested the sensitivity of aerosol concentrations at the surface to a strong

source of particles in the FT in simulation SOA_PRSC, to see if the particles transported into the FT could be entrained into the Arctic BL.

In the ASCOS region (the central Arctic), SOA_PRSC is the only simulation to produce 3–14 nm particle concentrations close to the observed profiles (Fig. 5). The nucleation mode concentration in SOA_PRSC is approximately an order of magni-tude greater than than of SOA, including at the surface, suggesting that FT entrainment of secondary particles is taking place

in the high Arctic in the model. In the ATom region over continental North America, nucleation mode concentrations from SOA_PRSC are greater than that of SOA in most of the FT, with the greatest difference being nearly an order of magnitude at 4 km. We prescribed the nucleation rate in the model for latitudes north of 80 °N, which is further north than the ATom area. We

also tested the prescribed rate in gridboxes north of 60 °N and still found good agreement with the ATom and ASCOS profiles (Figs. D1 and D2).

The higher nucleation mode concentrations at the surface in SOA_PRSC relative to SOA (Fig. 5) confirms that particles that have been created in the FT can be entrained into the Arctic boundary layer in the model. Since we prescribed a stronger FT NPF rate for all latitudes north of 80° in SOA_PRSC, the vertical transport of the secondary particles could be occurring in any location over such a region and then be transported through the BL to the location of the helicopter. In the next section, we will examine the effect of this FT source of particles on the particle size distribution at the surface during the AO2018 campaign.

The model output shown in Figs. 5 and 6 is for the year 2018, while the observational data is from 2008 (ASCOS) and 2016 (ATom). However, the large underestimations of the nucleation mode by simulation CONTROL (up to 3 orders of magnitude at the surface) for both campaigns is unlikely to be because of interannual variability alone. The CONTROL simulation also underestimates the particle concentrations from the AO2018 campaign, which was in the same year as our model output. We will show in the next section that simulation SOA_PRSC performs better than CONTROL when evaluated with AO2018

measurements, consistent with the conclusions of this section using older measurements.

### 4.3.2   Time series of particle concentrations

Time series and PDFs of particle number concentration at the surface from AO2018 observations and simulations CONTROL and SOA_PRSC are shown in Fig. 7 for nucleation, Aitken and accumulation mode particles.

In the nucleation mode, simulation SOA_PRSC increases the simulated particle concentration by 2–3 orders of magnitude

in the melt period relative to CONTROL (Fig. 7), and by an order of magnitude relative to SOA (Fig. 3). In the freeze period, the higher FT NPF rate in SOA_PRSC produces a nucleation mode concentration that is 1–2 orders of magnitude higher than SOA. This higher concentration in SOA_PRSC demonstrates that the FT is acting as a source of particles to the surface in the model. SOA_PRSC produces nucleation mode concentrations close to 100 cm$^{-3}$ in both the freeze and melt periods. As such, SOA_PRSC overestimates the nucleation mode concentration in the melt period and captures the lower end of the observed

distribution of nucleation concentration in the freeze period, but does not capture the highest concentrations seen during the NPF events.

In the Aitken mode, particle concentrations are also higher in SOA_PRSC than in CONTROL. Despite differences between SOA and SOA_PRSC in the nucleation mode, they produce the same concentration of Aitken particles. Melt-period Aitken particle concentrations in SOA_PRSC are 1–3 orders of magnitude higher than in CONTROL, taking them to within 1 order of

magnitude of the observed particle concentration for most of the melt period. As in the nucleation mode, Aitken concentrations decrease in SOA_PRSC in the freeze period such that the observations are underestimated by 1–2 orders of magnitude in the freeze period. The declining Aitken concentrations suggests a decrease in the particle growth rates towards late summer, possibly driven by declining photochemical production of precursor vapours (see appendix E). The Aitken mode concentration decreases despite the fixed NPF rate above the BL in SOA_PRSC, highlighting that particle growth by condensation is an

important process as well as the particle formation rate itself.

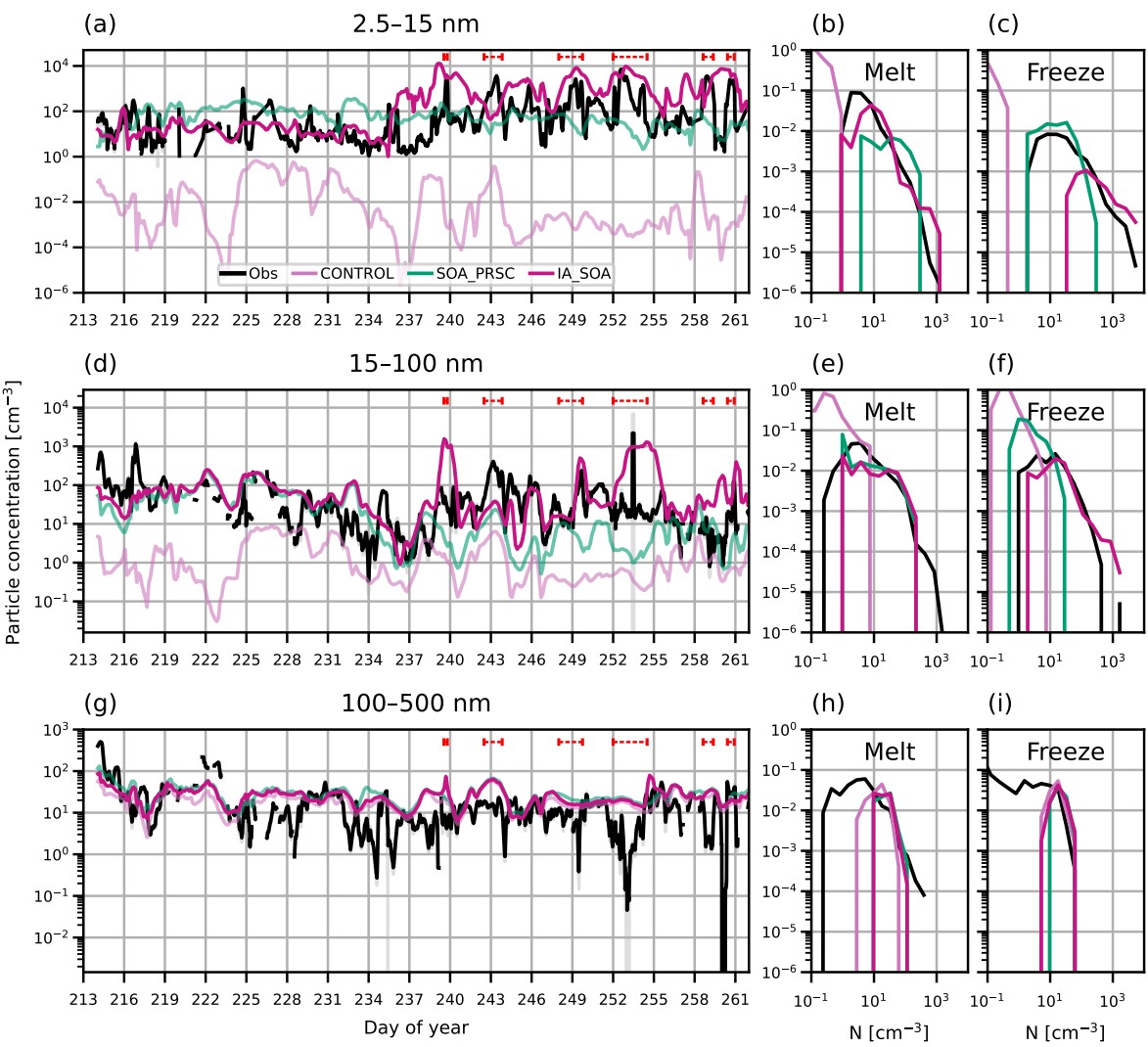

**Figure 7.** Time series and PDFs of aerosol concentration at the surface during AO2018 from observations and model output. Model output is from simulations CONTROL (pink lines), SOA_PRSC (green lines) and IA_SOA (red lines). Other than IA_SOA and the observations, lines have been made slightly transparent on this figure so that it is easier to view the results from IA_SOA. Observations are shown as 3-hourly mean (black lines) and standard deviation (grey shading). Aerosol concentrations are shown for particles with diameter (a–c) 2.5–15 nm, (d–f) 15–100 nm and (g–i) 100–500 nm. Red dashed lines in (a, d, g) show observed NPF events. PDFs are separated by observed sea ice freeze-up date, 27th August 2018 (day 239).

The accumulation mode is well captured by SOA_PRSC, like in SOA and CONTROL. As for IA, the changes to the model NPF scheme in the SOA simulations have little effect on the accumulation mode concentration relative to CONTROL.

| Vapour species | Change in concentration from Aug to Sep 2018 [%] |
|---|---|
| DMS | -28.8 |
| $SO_2$ | 135.2 |
| $H_2SO_4$ | -85.2 |
| OH | -87.7 |
| Monoterpenes | 31.2 |
| Secondary organics | -5.9 |

**Table 4.** Table of change in simulated concentrations of different vapours from August to September 2018 in simulation CONTROL. Concentration changes are given as the change in the mean value over gridboxes at the surface with latitude 80–90 N.

## 4.4 Combining local and non-local NPF

Overall, SOA performs well in the melt period of AO2018 but still underestimates particle concentration in the freeze period, when simulation IA performs well. This suggests that a combination of transported secondary particles and local BL NPF involving iodic acid is required to capture the full aerosol time series. To test this, we combined the use of the organically-mediated NPF scheme with our iodic acid NPF scheme. The combined simulation of SOA and IA is shown as simulation IA_SOA in Fig. 7. IA_SOA captures both the baseline of the nucleation mode in the melt period and the peaks in the freeze period. In the Aitken mode, IA_SOA is within 1 order of magnitude of the observations except for a few brief periods, such as days 253–255 when the observed concentration is overestimated in the model.

The behaviour of the simulations can be summarised by comparing the overlap of PDFs of simulated particle concentration with those from observations (Table 3). Simulation SOA has higher overlap indices for the nucleation (0.73) and Aitken modes (0.73) than CONTROL in the melt period, when the source of secondary particles from the FT has the greatest influence, but shows less of an improvement in the freeze period. IA_SOA also shows improvement relative to CONTROL in the melt period. During freeze-up, IA_SOA slightly overestimates the nucleation mode concentration, meaning that its nucleation mode overlap index in this period (0.51) is lower than that of SOA_PRSC (0.52), which does not overestimate the concentration in the same way. However, it is clear from the time series that the simulations with iodic acid NPF (IA, IA_SOA) capture the nucleation concentration peaks from NPF events in a way that the SOA_PRSC simulation does not. Simulation IA_SOA has the highest overlap index of all simulations for the Aitken mode in the freeze period (0.77).

## 5 Discussion and conclusion

We have used field observations and a global aerosol-climate model with an empirical iodic acid nucleation scheme to investigate sources of aerosol in the high Arctic summer. Our results point to a regime transition occurring in late summer from a free-tropospheric source of secondary particles to in situ new particle formation in the boundary layer, driven by iodic acid. The onset of iodic acid new particle formation (triggered by sea ice freeze-up) coincides with a decline in the free tropospheric

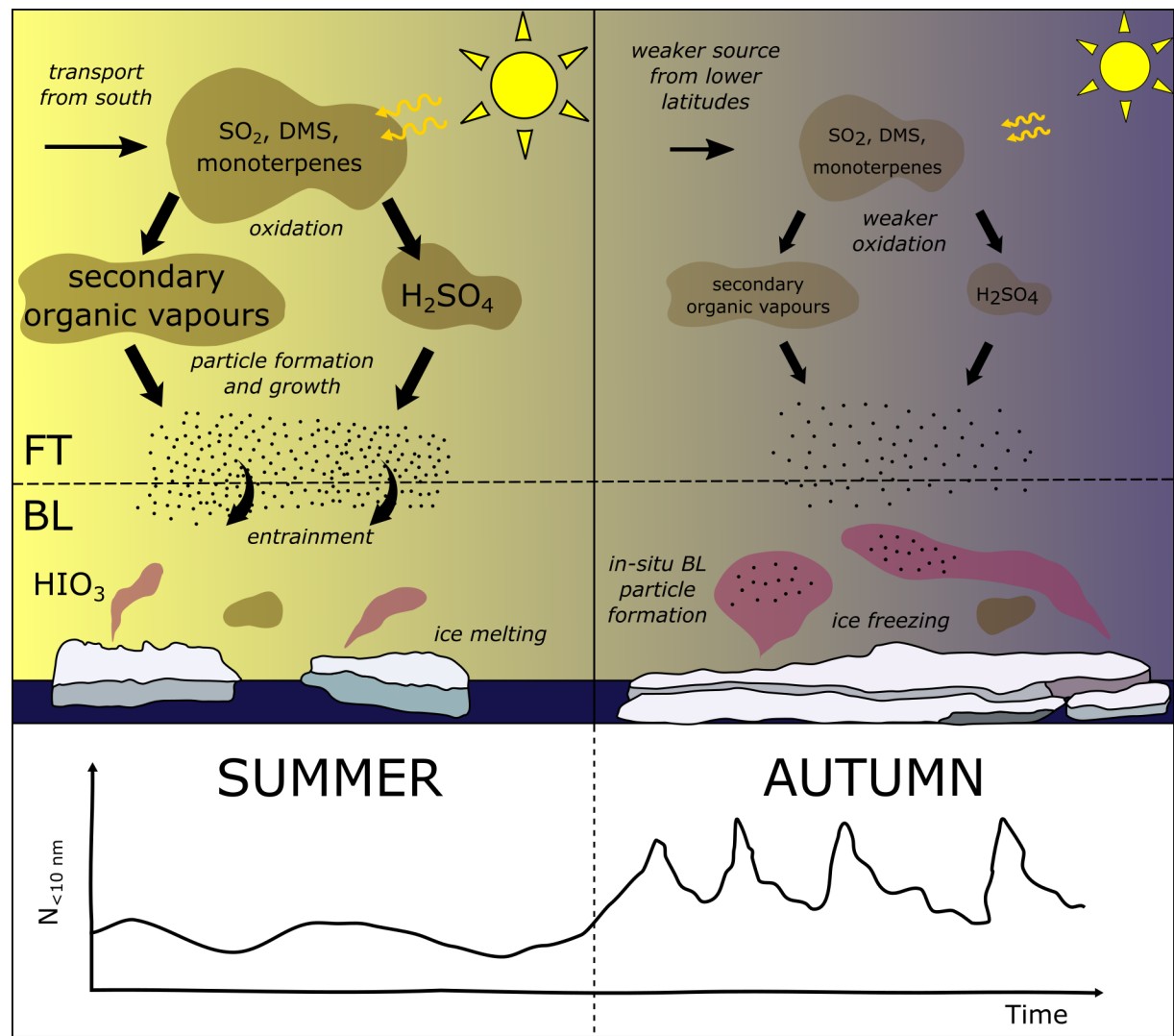

**Figure 8.** Schematic of processes controlling the concentration of nucleation mode particles at the surface in the high Arctic. The summer sea ice melt period is on the left and the freeze-up period in late summer/early autumn is on the right. A schematic of the nucleation mode particle concentration is shown at the bottom.

source rate brought on by declining rates of photochemical production of precursor vapours in the free troposphere. The net effect of the transition from free troposphere to boundary layer nucleation is a fairly constant nucleation and Aitken mode at the surface.

There are several key conclusions we can draw from the simulations we have presented here. They are listed below.

– The default settings of the UKESM1 model cannot capture Arctic aerosol concentrations. The nucleation mode particle concentration at the surface is at least an order of magnitude too low in CONTROL compared to AO2018, ASCOS

and ATom observations, sometimes underestimating observed concentrations by as much as 5 orders of magnitude. The surface Aitken mode concentration is also underestimated by up to 3 orders of magnitude. The model performs better at simulating the surface accumulation mode concentration. The accumulation mode concentration in CONTROL is generally within the range of the observations, but does not capture the lowest concentrations (less than 1 cm$^{-3}$) seen in the time series of AO2018 observations. These periods of low accumulation mode concentration are important in controlling the behaviour and radiative effects of low-level Arctic clouds (Mauritsen et al., 2011; Loewe et al., 2017; Birch et al., 2012; Stevens et al., 2018).

– Our simulation with new particle formation in the boundary layer produces more accurate nucleation and Aitken mode concentrations at the surface during the AO2018 melt period than the CONTROL simulation. However, modelled aerosol concentrations aloft, combined with results of a simulation with boundary layer new particle formation north of 85°N, shows that little to no boundary layer new particle formation takes place in situ during the AO2018 before the iodic acid new particle formation events occur. This result is in contrast to results from Browse et al. (2014), who found that new particle formation from $H_2SO_4$ in the Arctic boundary layer was an important source of high-Arctic CCN in GLOMAP. Instead, in these UKESM simulations, nucleation and Aitken mode particles at the surface in the high Arctic were created at lower latitudes and then transported northwards through the free troposphere. Higher concentrations at the surface from a simulation with a prescribed new particle formation rate in the low free troposphere supports the hypothesis of a free-tropospheric source to the surface (Fig. 8, left-hand side).

– Our simulations suggest that a seasonal regime shift triggered by changes in photochemistry coincides with the beginning of the iodic acid season triggered by the sea ice freeze-up. This is portrayed in Fig. 8. The net effect of these two changes is a fairly constant source of particles controlled by new particle formation from two different mechanisms. Photochemistry declines towards the end of the AO2018 campaign period as a result of the reduction in incoming solar radiation at the end of summer. In the model output from simulation CONTROL, surface concentrations of the OH radical decline by 87.7% over 80–90°N from August 2018, causing $H_2SO_4$ to decline by 85.2% while $SO_2$ increases in concentration in the region by 135.2% over the same period (Table 4). The reduction in $H_2SO_4$ inhibits new particle formation and particle growth (Fig. E5). In simulation SOA_PRSC, the Aitken mode particle concentration declines from August to September 2018 even though the new particle formation rate in the free troposphere is prescribed to be constant. This behaviour highlights the importance of particle growth rates, which can vary alongside the new particle formation rate itself. Following this decline in the extra-Arctic aerosol source, the local iodic acid-driven particle formation begins.

Our results have implications for the future Arctic aerosol budget. Iodic acid new particle formation in the Arctic boundary layer is strongly coupled to the surface and therefore sensitive to changes in the sea ice. However, our simulations show that entrainment of secondary particles from the free troposphere is also an important source of surface aerosol at nucleation and Aitken mode sizes. Free-tropospheric new particle formation is unlikely to have such a strong sensitivity to local sea ice changes, since the precursor vapours and background aerosol in the free troposphere are likely to have been transported from other regions. Further work will be required to understand how the balance of boundary layer versus free-tropospheric nucle-

580 ation will evolve in the changing Arctic, since predictions of cloud behaviour and surface energy balance in the future Arctic depend on knowledge of the Arctic aerosol budget. Previous studies have shown that models struggle to simulate the thermo-dynamic structure of the boundary layer in the high Arctic, for example by failing to produce the decoupled conditions that are common (Birch et al., 2012; Sotiropoulou et al., 2016). Such biases could inhibit accurate predictions of free-tropospheric aerosol sources, since the entrainment of particles in the boundary layer will rely on accurate representation of the turbulent

mixing created by the structure of the cloud layer.

This study shows that it is important to assess the influence of boundary layer versus free-tropospheric sources compared to other uncertain Arctic aerosol processes and their different representations in models, for example sea spray parameterisations and primary marine sources. We did not consider primary sources for the Aitken mode during AO2018, such as biogenic marine particles. Although our model includes primary marine aerosol emissions, these are mostly limited to the accumulation mode.

Thus, our model does not include direct marine emissions of smaller particles. Previous field studies in the Arctic pack ice region have found evidence that organics in Aitken and accumulation mode aerosols and cloud water were related to polymer gels found in the surface microlayer of sea water, suggesting a primary marine aerosol source (Bigg et al., 2001; Leck and Bigg, 2005; Bigg and Leck, 2008; Leck and Bigg, 2010; Orellana et al., 2011; Karl et al., 2013; Hamacher-Barth et al., 2016), although our earlier modelling results suggest that to account for the observed Aitken mode concentrations would require

an unrealistically high surface source (Korhonen et al., 2008b). Also proposed in the literature is an atmospheric processing pathway where larger primary particles break-up to form more, smaller particles. We cannot use our model results to exclude such a source from the aerosol size distributions measured during AO2018. The balance of primary and secondary aerosol sources in this region merits further work, and would likely require improvements to existing model parameterisations (i.e. new particle formation and growth rates, primary marine emission fluxes) and the development of new parameterisations to

test in the model (i.e. particle emissions from open leads independent of wind speed, and size-resolved break-up rates for primary marine particles).

A greater understanding of aerosol conditions in the Arctic free troposphere is needed. It is not possible to distinguish from the model output presented here whether particles coming from the free troposphere have been created there or have first been transported from lower latitudes, where they were created in the boundary layer. Our results indicate the importance of

605 obtaining measurements of aerosol size distributions and precursor vapours from above the boundary layer in the high Arctic, which have previously been sparse. Moreover, the Kulmala et al. (2006) $H_2SO_4$ and Metzger et al. (2010) organically-mediated new particle formation schemes we use produce very similar surface particle concentrations in the region of study, such that we cannot use these datasets to evaluate the accuracy of one over the other. This highlights an open question in modelling of Arctic aerosol. It is crucial to understand which precursor species are important for Arctic aerosol in order to understand how

changes to different parts of the climate system will affect the formation and behaviour of Arctic clouds. The production of secondary organic vapour in UKESM1 is crude, accounting only for the oxidation of monoterpenes, and we have not included any effects of ammonia, which has been shown in laboratory and Arctic field studies to contribute to new particle formation. The inclusion of ammonia in the model could lead to more new particle formation in the boundary layer since ammonia can

stabilise H$_2$SO$_4$ clusters. However, before these potential sources of model bias can be improved, more observational data is needed about what drives new particle formation throughout the Arctic, both in the free troposphere and at the surface.

The accumulation mode in the model is relatively unaffected by the different new particle formation schemes we used, and is better simulated than the smaller modes. This shows that accumulation mode aerosol and the contribution of the particles to CCN is not affected by new particle formation in these simulations. However, comparison with AO2018 observations shows that the model cannot capture periods of up to a day of very low accumulation mode concentration (approximately 1 cm$^{-3}$ and below). Given that the accumulation mode affects new particle formation via the sink of condensable vapours, it will be important to consider this model bias in future work. Moreover, observations indicate the importance of fog as a sink of the iodic acid, so more work is needed to assess the role of fog in controlling the frequency of iodic acid new particle formation events. A higher resolution model with better cloud parameterisations would be better suited to such studies than the coarse, global model we have used here.

*Code availability.* The code used to analyse model output and produce the figures is available on Github at https://github.com/ruthprice/price-acp-figures.

*Data availability.* Data from the Arctic Ocean 2018 campaign is available on the Bolin Data Centre, https://bolin.su.se/data/oden-ao-2018-expedition-2. The data used here from the ASCOS campaign is available on the Bolin Centre Database, https://doi.org/10.17043/oden-ascos-2008-aerosol-stratification-1. All data from the Atmospheric Tomography mission are openly available and archived in the Oak Ridge National Laboratory Distributed Active Archive Center (ORNL DAAC) at https://doi.org/10.3334/ORNLDAAC/1925.

## Appendix A: Surface temperature during AO2018

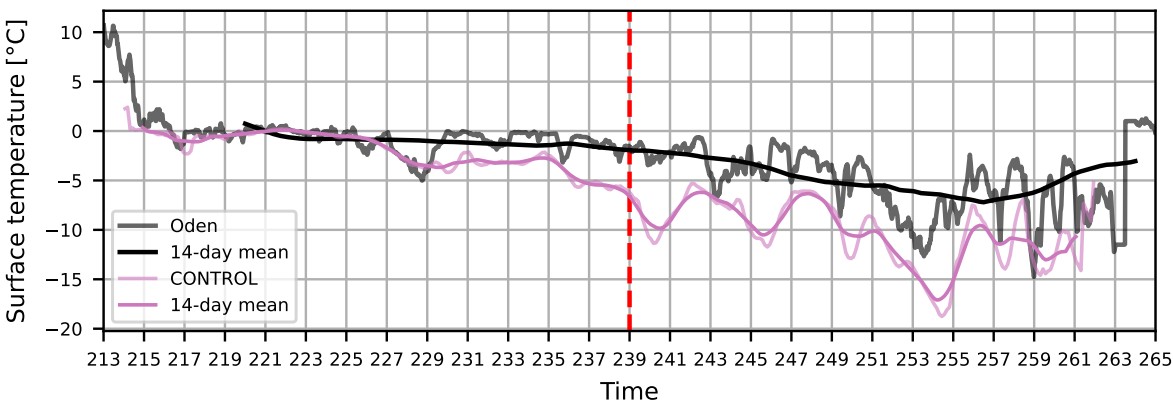

**Figure A1.** Time series of surface temperature during the AO2018 campaign periods from observations and simulation CONTROL. Observations are shown as 30-min mean values (grey line) and 14-day running mean values (black line). Model output is shown as 3-hourly mean values (pale pink line) and 14-day running mean values (dark pink line). The observed sea ice freeze-up point (when the 14-day mean reaches -2 °C) is marked with a red dashed line.

Figure A1 shows the surface temperature during AO2018 from ship measurements and CONTROL output. The sea ice freeze-up point is defined to be when the 14-day running mean of the surface temperature reaches -2 °C. During AO2018, this occurred on 27th August 2018 (day 239). The model output at this time gives a surface temperature of approximately -6 °C. We use -5 °C as the proxy for sea ice freezing in the iodic acid scheme instead of -2 °C, so that the iodic acid is triggered on roughly the observed freeze-up day in the model.

## Appendix B: Choice of new particle formation parameterisation

In section 4 we show that switching on NPF in the BL in the model creates more particles at the surface than using the Vehkamäki et al. (2002) scheme only (simulation SOA relative to CONTROL). To test the sensitivity of the results to the choice of BL NPF scheme, we ran simulations using the K06 scheme (simulation SA) and using the M10 scheme in the FT as well as the BL (SOA_ALL_LEVELS). Aerosol number concentrations at the surface from these simulations are shown in figure B1. Interestingly, simulations SA, SOA and SOA_ALL_LEVELS have very similar aerosols concentrations at the surface, despite the differences in the NPF schemes. SA occasionally simulates higher nucleation mode concentrations than SOA or SOA_ALL_LEVELS, most notably on days 239–243.

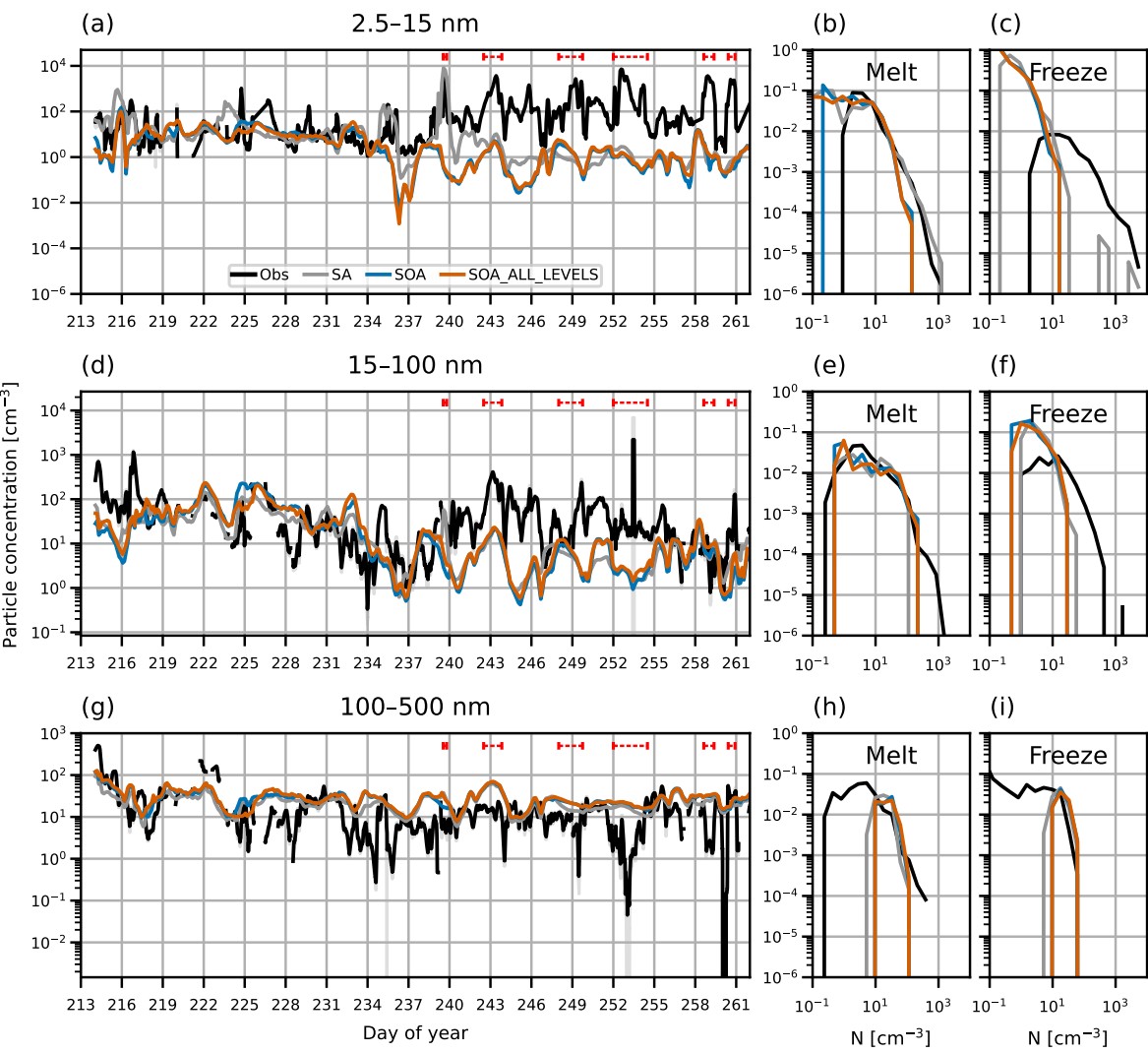

**Figure B1.** Time series and PDFs of surface aerosol concentration during AO2018 from observations and model output. Model output is from simulations SA (grey lines), SOA (blue lines) and SOA_ALL_LEVELS (orange lines). Observations are shown as 3-hourly mean (black lines) and standard deviation (grey shading). Aerosol concentrations are shown for particles with diameter (a–c) 2.5–15 nm, (d–f) 15–100 nm and (g–i) 100–500 nm. Red dashed lines in (a, d, g) show observed NPF events. PDFs are separated by observed sea ice freeze-up date, 27th August 2018 (day 239).

 **Appendix C: Particle ageing**

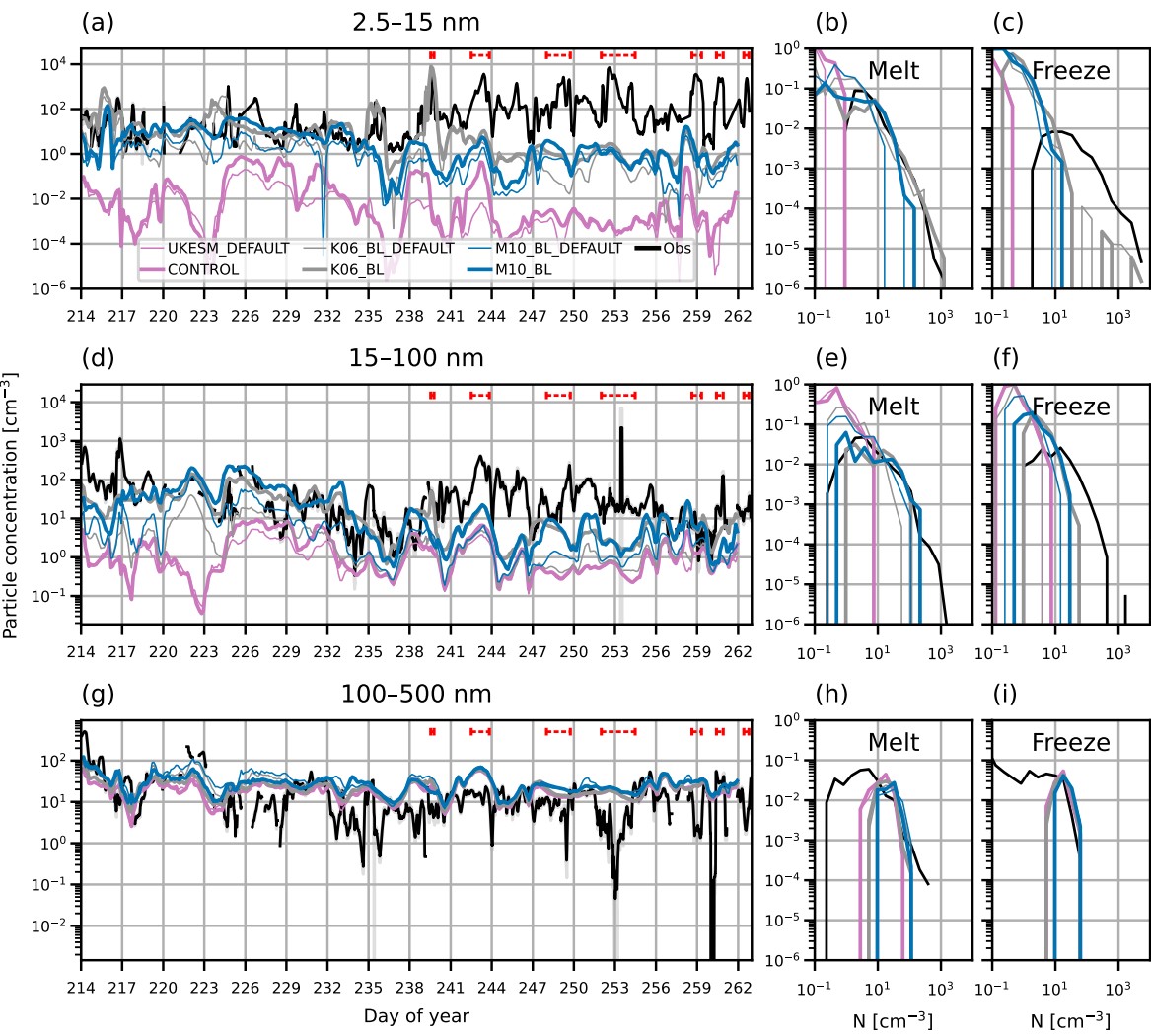

**Figure C1.** Time series and PDFs of surface aerosol concentration during AO2018 from observations and model output. Model output is from simulations UKESM_DEFAULT (thin pink lines), CONTROL (thick pink lines), SA_DEFAULT (think grey lines), SA (thick grey lines), SOA_DEFAULT (thin blue lines) and SOA (thick blue lines). Observations are shown as 3-hourly mean (black lines) and standard deviation (grey shading). Aerosol concentrations are shown for particles with diameter (a–c) 2.5–15 nm, (d–f) 15–100 nm and (g–i) 100–500 nm. Red dashed lines in (a, d, g) show observed NPF events. PDFs are separated by observed sea ice freeze-up date, 27th August 2018 (day 239).

In Sect. 4.2 we show that SA and SOA simulate more accurate concentrations of the nucleation and Aitken modes during AO2018 than simulation CONTROL (3). Figure C1 shows the surface aerosol concentrations for simulations UKESM_default

and CONTROL, SA_default and SA, and SOA_default and SOA. The simulations with the ageing change produce higher concentrations in the nucleation and Aitken modes than the DEFAULT simulations, and this difference is greater in the simulations with BL NPF than it is between UKESM_DEFAULT and CONTROL. This is because the mode merging described in Sect. 3.4 affects the Aitken mode concentration more when there is a stronger source of smaller particles. In the Aitken mode, the concentrations from simulations SA and SOA are typically higher than SA_DEFAULT and SOA_DEFAULT by a factor of 2–10. The model-observation agreement is therefore improved in simulations with the ageing change.

Figure C2 shows mean aerosol profiles simulated for the AO2018 period for the soluble nucleation, Aitken and accumulation modes and the insoluble mode. Model output is shown for simulations CONTROL, UKESM_default, SA(_default) SOA(_default), SOA_ALL_LEVELS(_default), SOA_ALL_LEVELS(_default) and SOA_PRSC(_default). The perturbed ageing scheme produces up to an order of magnitude more Aitken mode particles than the default scheme in the low FT and at the surface, independently of which NPF scheme is used. The insoluble mode is not significantly affected by the changes to the ageing scheme.

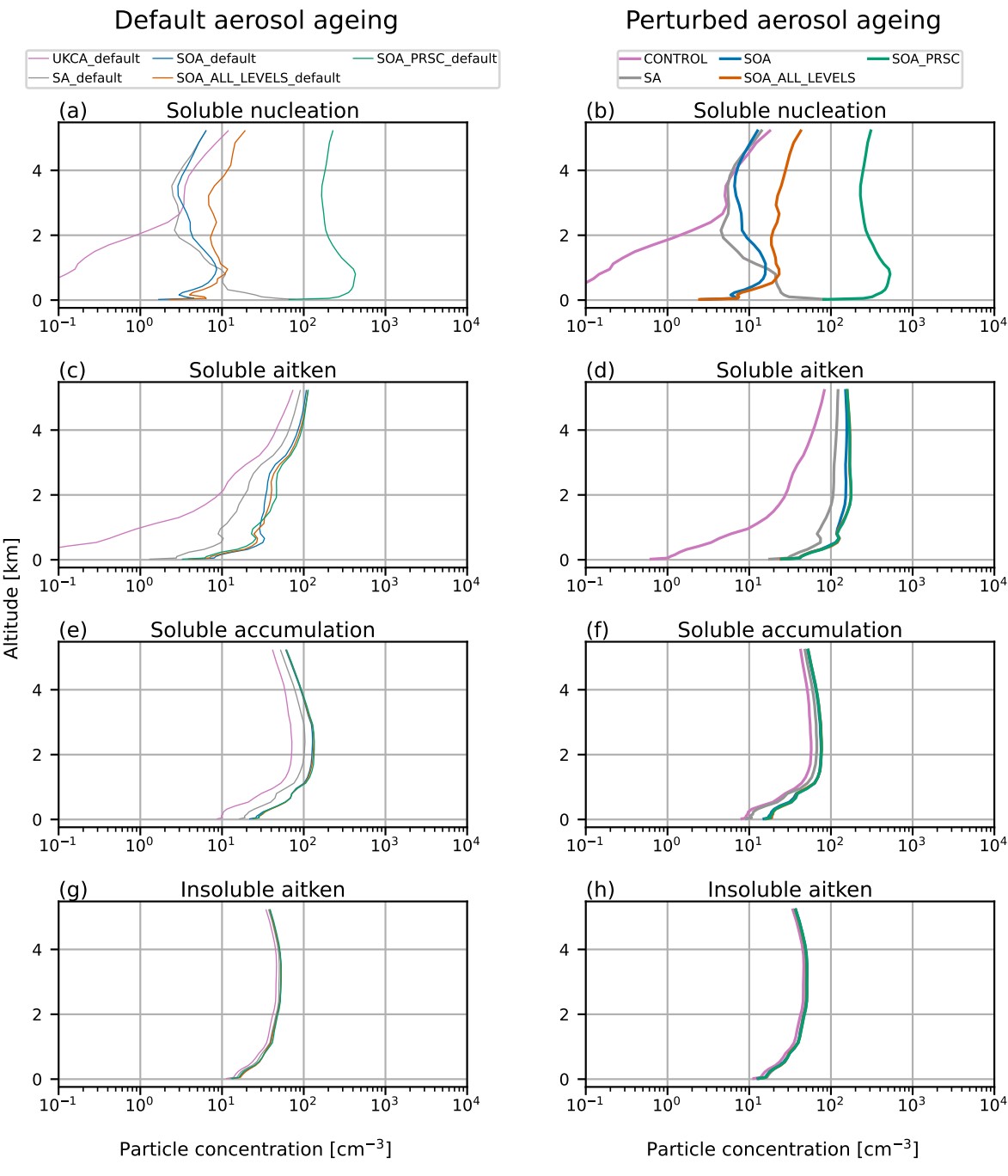

**Figure C2.** Mean simulated aerosol profiles for AO2018 campaign period. Concentrations shown are for (a–b) soluble nucleation, (c–d) soluble Aitken, (e–f) soluble accumulation and (g–h) insoluble Aitken modes. Model output is from simulations UKCA_DEFAULT and CONTROL (pink lines), SA_default and SA (grey lines), SOA_ALL_LEVELS_default and SOA_ALL_LEVELS (orange lines), SOA_default and SOA (blue lines) and SOA_PRSC_default and SOA_PRSC (green lines).

## Appendix D: Latitude limit for prescribed FT NPF rate

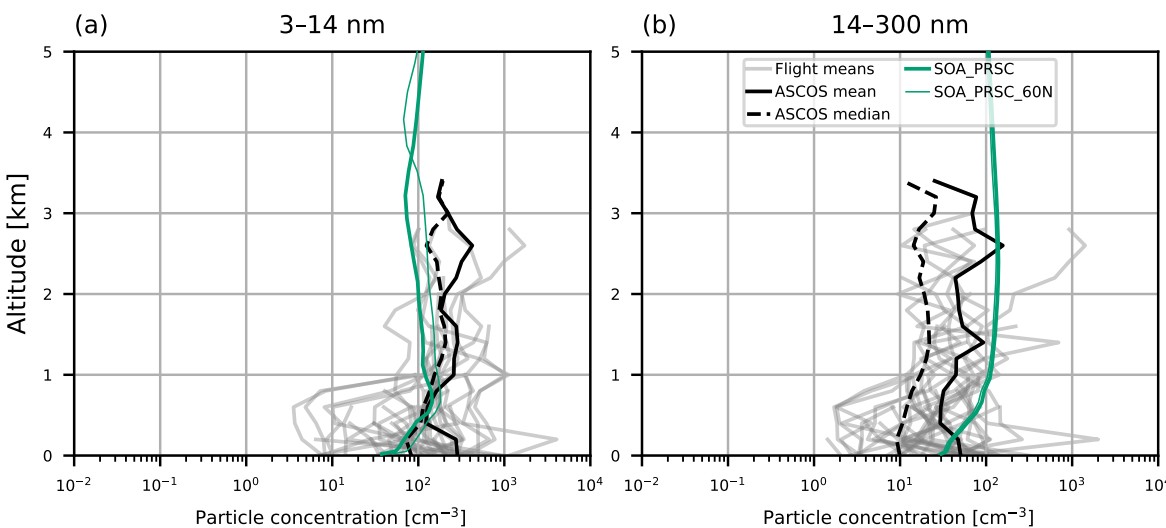

**Figure D1.** Aerosol vertical profiles from model output and ASCOS campaign observations. Model output is from colocated monthly mean values from simulations SOA_PRSC (thick dark green) and SOA_PRSC_60N (thin dark green). Observed values are given as the mean profile from each ASCOS flight (grey lines), overall mean (solid black) and overall median (dashed black). Profiles are for particles with size (a) 3–14 nm, measured during ASCOS using a UCPC and (b) 14–300 nm, measured using a CPC (particles greater than 14 nm) and a CLASP instrument (particles greater than 300 nm).

In simulation SOA_PRSC, we impose a NPF rate of $10^{-2}$ cm$^{-3}$ s$^{-1}$ in model levels between the top of the boundary layer and 7.5 km altitude. The rate is imposed in gridboxes north of 80°N. In Sect. 4.3.1 we compare the SOA_PRSC output to aerosol profiles from the ASCOS and ATom campaigns to test how realistic such a NPF rate may be. However, the data we use from ATom was taken further south, between approximately 60–80°N (see Fig. 1). To test our prescribed FT NPF rate using ATom observations we therefore ran another simulation, SOA_PRSC_60N, where the rate is imposed north of 60 °N.

Figures D1 and D2 show output from SOA_PRSC and SOA_PRSC_60N with observations from ASCOS and ATom. Aerosol concentrations from the two simulations are within an order of magnitude of each other. While the prescribed rate increases aerosol concentrations relative to CONTROL in the ASCOS region (Sect. 4.3.1), it makes little difference in the ATom region even when we extend the region in which the rate is applied. This is likely because NPF rates are already higher in the ATom region than the ASCOS region due to the relative proximity of ATom to open water and boreal forests, both of which supply precursor vapours to the atmosphere.

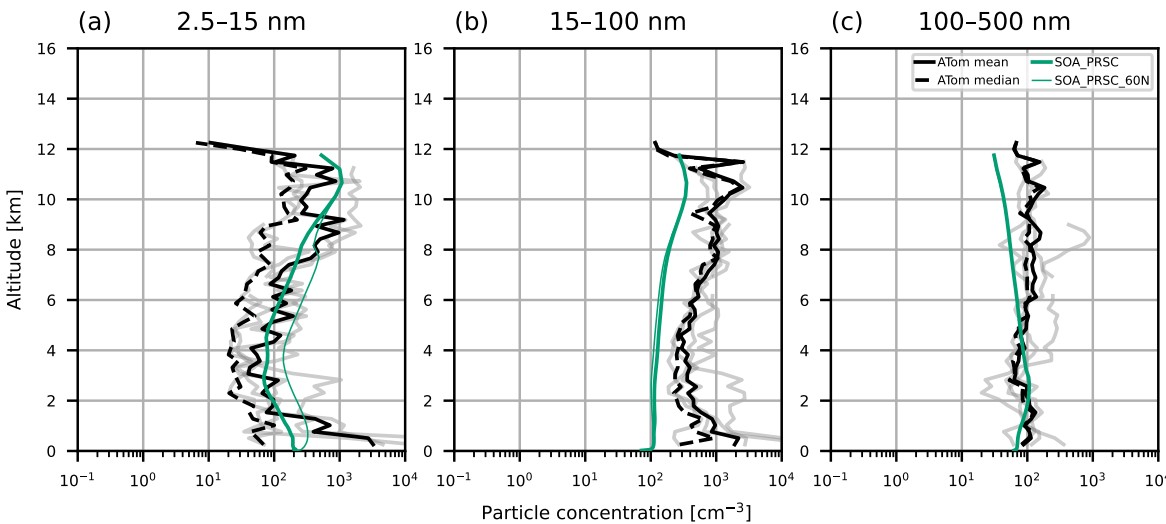

**Figure D2.** Aerosol vertical profiles from model output and ATom campaign observations. Model output is from colocated monthly mean values from simulations SOA_PRSC (thick dark green) and SOA_PRSC_60N (thin dark green). ATom observations are taken from leg 1 of the campaign and restricted to measurements that were taken north of $60°$N. Observations correspond to mean profiles from different days (grey lines), the overall mean (black solid lines) and overall median (black dashed lines). Profiles are for particles with size (a) 5–10 nm, (b) 10–100 nm and (c) 100–500 nm. Observations were recorded at standard temperature and pressure, model output has been adjusted to account for this.

## Appendix E:  Aerosol precursor vapours and growth rate

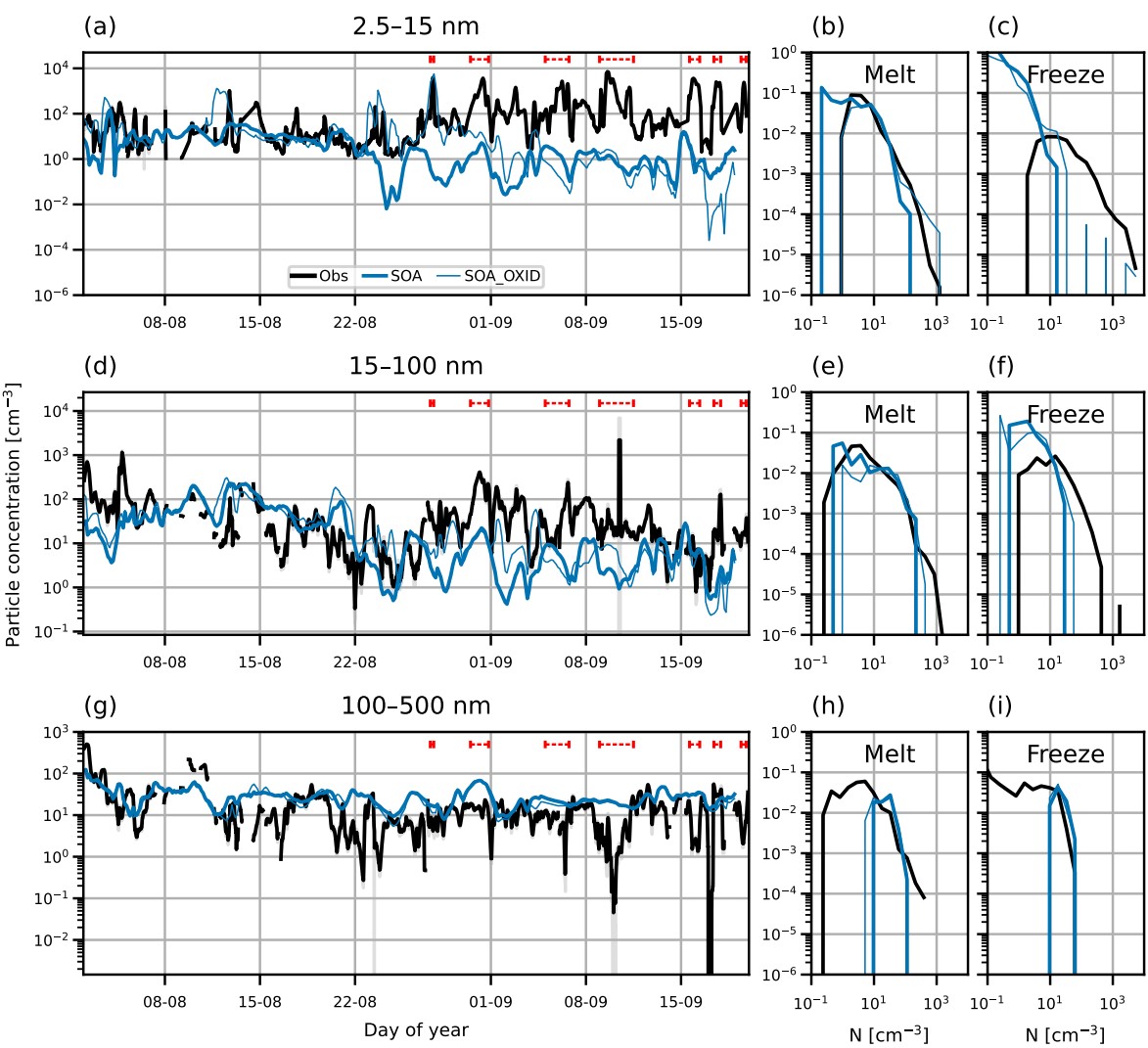

**Figure E1.** Time series and PDFs of surface aerosol concentration during AO2018 from observations and model output. Model output is from simulations SOA (thick blue lines) and SOA_OXID (thin blue lines). Observations are shown as 3-hourly mean (black lines) and standard deviation (grey shading). Aerosol concentrations are shown for particles with diameter (a–c) 2.5–15 nm, (d–f) 15–100 nm and (g–i) 100–500 nm. Red dashed lines in (a, d, g) show observed NPF events. PDFs are separated by observed sea ice freeze-up date, 27th August 2018 (day 239).

In our simulations we have tested the effect of different NPF rates on the modelled aerosol concentrations. However, particle formation is also affected by the availability of condensable vapours (if new particles grow faster, they are more likely to

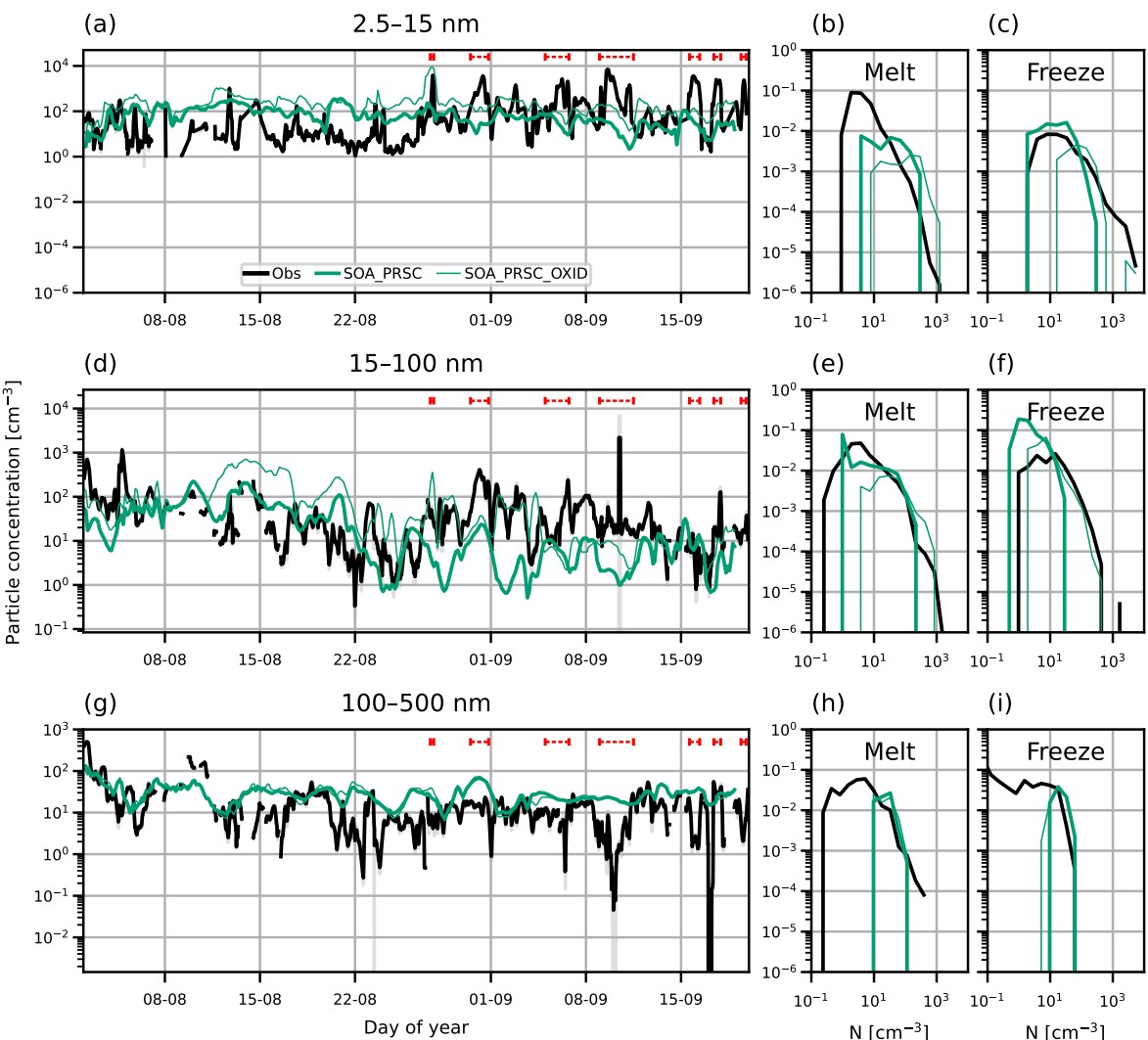

**Figure E2.** Time series and PDFs of surface aerosol concentration during AO2018 from observations and model output. Model output is from simulations SOA_PRSC (thick greem lines) and SOA_PRSC_OXID (thin green lines). Observations are shown as 3-hourly mean (black lines) and standard deviation (grey shading). Aerosol concentrations are shown for particles with diameter (a–c) 2.5–15 nm, (d–f) 15–100 nm and (g–i) 100–500 nm. Red dashed lines in (a, d, g) show observed NPF events. PDFs are separated by observed sea ice freeze-up date, 27th August 2018 (day 239).

survive in the atmosphere for longer rather than being lost to coagulation with other particles). Therefore, since growth is important, we tested the sensitivity of NPF in the SOA and SOA_PRSC simulations to the availability of secondary organic vapour by allowing more vapour to be transported to the Arctic in simulations SOA_OXID and SOA_PRSC_OXID. Time series and PDFs of surface particle concentration during AO2018 are shown on Fig. E1 and Fig. E2. In the nucleation mode,

SOA_PRSC_OXID produces a concentration of approximately $100 \text{ cm}^{-3}$, behaving similarly to SOA_PRSC. SOA_OXID

produces nucleation mode concentrations up to 3 orders of magnitude greater than SOA for brief periods between 22nd August and 1st September 2018, but overall this effect is not enough to account for the underestimation of the observed nucleation mode concentrations by SOA in the freeze period.

In the Aitken mode, there are periods where SOA_PRSC_OXID behaves the same as SOA_PRSC and brief periods where it produces higher concentrations. For example, on days 223–229, the concentration in SOA_PRSC_OXID is nearly an order of

magnitude greater than that from SOA_PRSC, resulting in an overestimation of the observed Aitken concentration. The Aitken mode concentrations in SOA_OXID are similar to that of SOA.

Figures E3 and E4 show maps and zonal means of $H_2SO_4$ and secondary organic vapour concentration from simulation CONTROL. In August, Arctic $H_2SO_4$ concentration peaks just above the surface and at approximately 8 km (Fig. E3(a)). The concentration decreases throughout most of the troposphere from August to September by at least an order of magnitude. In

August, the secondary organics have a maximum at the surface from 60–70°N and some of this plume spreads north to the high Arctic. Unlike $H_2SO_4$, concentrations do not significantly reduce from August to September. The mean surface concentration of $H_2SO_4$ for 80–90°N reduces by 85.2% from August to September, while for secondary organics the reduction is only 5.9%. Percentage changes for other vapours are given in Table 4.

The decline in vapour concentration from August to September drives a reduction in the growth rate of aerosols from con-

densation of vapour. Figure E5 shows aerosol growth rates calculated from model output from simulation CONTROL. Arctic growth rates are lower than in the mid-latitudes and tropics, where $H_2SO_4$ and secondary organics have higher concentrations. In the central Arctic, the decreasing growth rate is driven by the decrease in $H_2SO_4$. Secondary organic vapours contribute little to the growth rate in the central Arctic region, but dominate in the continental Arctic in North America and northern Eurasia, where concentrations are high near the boreal forest source regions.

In Sect. 4.3.2 we showed that the source of aerosols from FT NPF weakens towards the end of summer. This occurs even when the FT NPF rate is held constant in time as in simulation SOA_PRSC. The slower aerosol growth rate shown in Fig. E5 curbs FT NPF in September. This highlights that it is important to understand the growth of new particles as well as their formation.

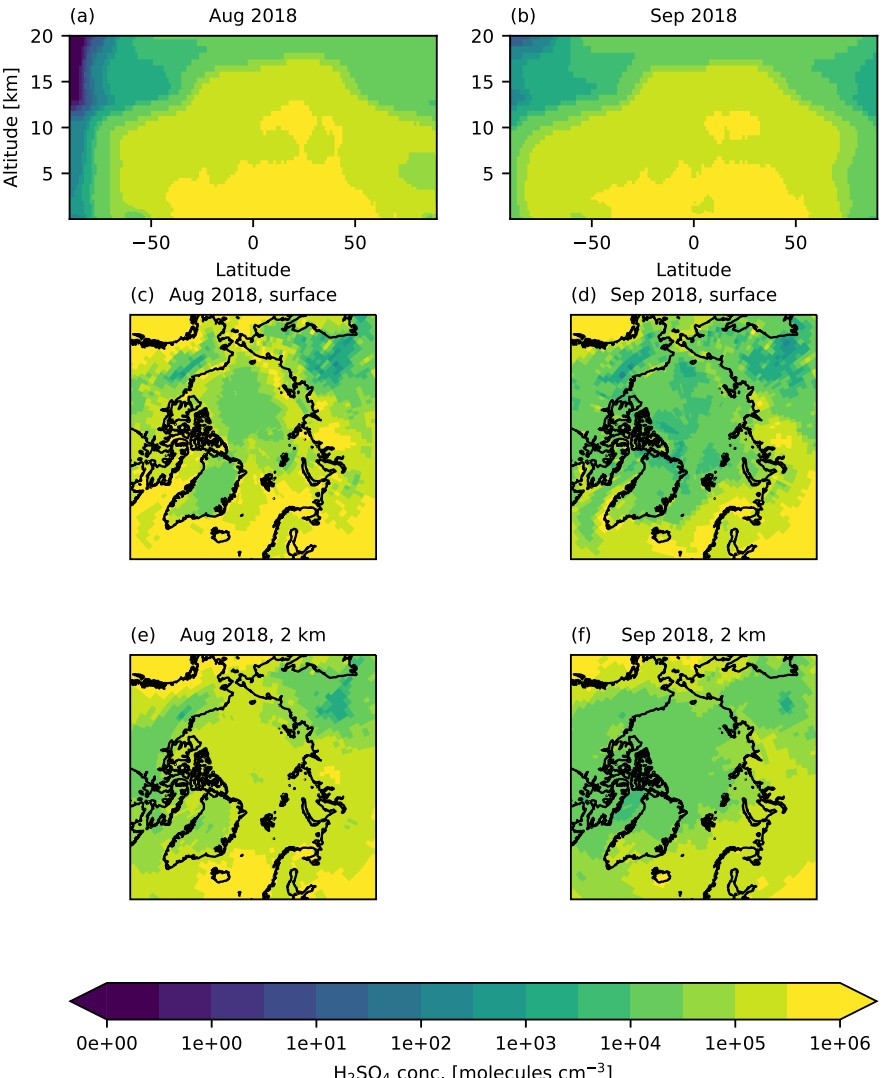

**Figure E3.** Zonal means and maps of simulated monthly mean H$_2$SO$_4$ concentration from simulation CONTROL. Maps are taken from model level (c–d) at surface and (e–f) with altitude 2 km.

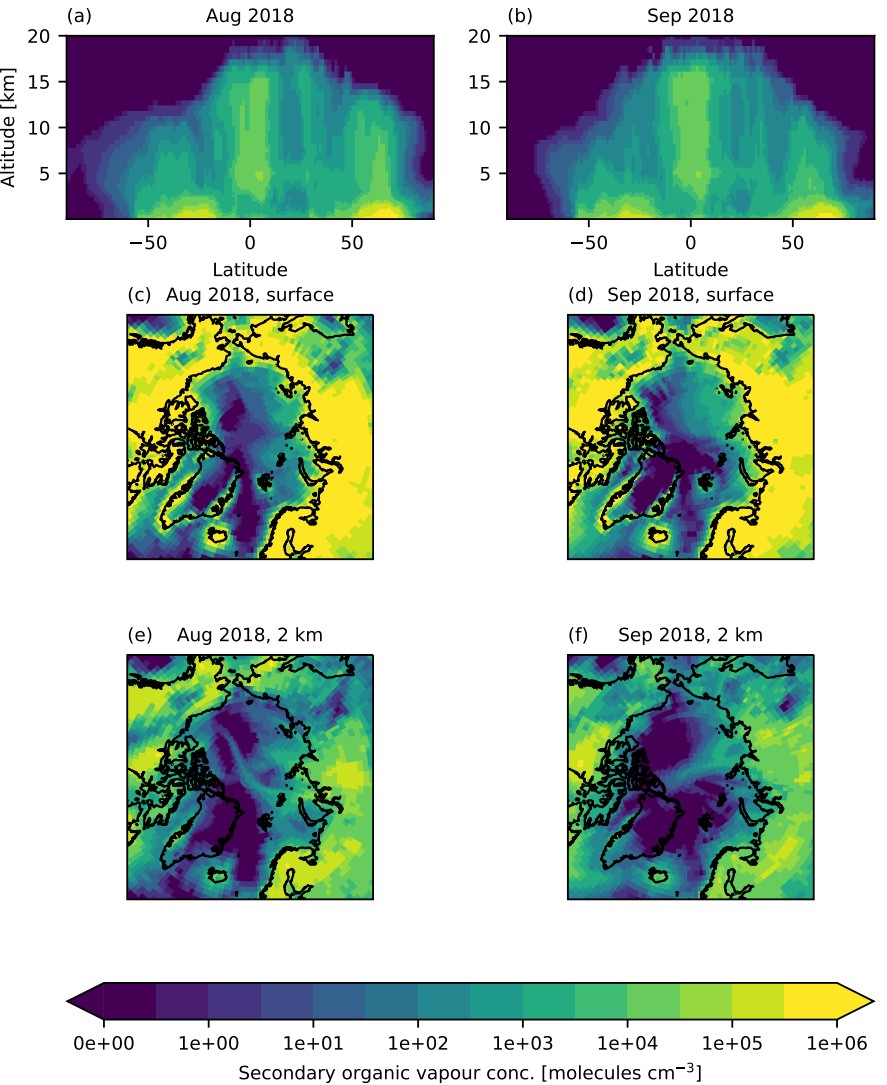

**Figure E4.** Zonal means and maps of simulated monthly mean secondary organic vapour concentration from simulation CONTROL. Maps are taken from model level (c–d) sat surface and (e–f) with altitude 2 km.

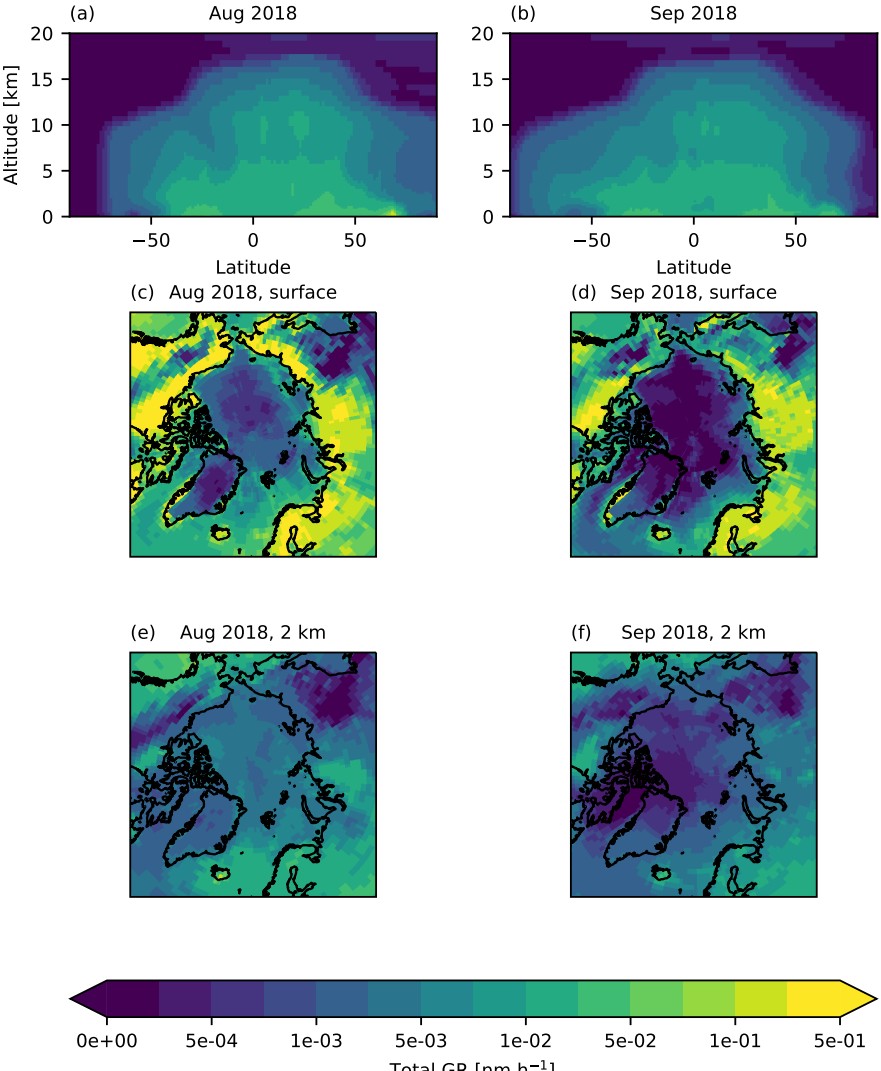

**Figure E5.** Zonal means and maps of aerosol growth rates from simulation CONTROL. Growth rates are calculated offline using model output of temperature and concentrations of $H_2SO_4$ and secondary organic vapour. Maps are taken from model level (c–d) at surface and (e–f) with altitude 2 km.

*Author contributions.* RP ran model simulations, assisted by PF, and analysed output with guidance from KSC. JS, AB and PZ performed aerosol measurements during AO2018. RP, PF, IMB and KSC contributed towards the design of the study. All authors were involved in discussion of results and writing of the manuscript.

*Competing interests.* KSC is an executive editor of ACP. The peer-review process was guided by an independent editor, and the authors have also no other competing interests to declare.

*Acknowledgements.* RP was supported by a Natural Environment Research Council (NERC) Ph.D. studentship through the SPHERES Doctoral Training Partnership (NE/L002574/1). PZ was supported by the Swedish Research Council (VR starting grant 2018-05045). JS and AB received funding from the Swiss National Science Foundation (Grant no. 169090) and the Swiss Polar Institute. JS holds the Ingvar Kamprad Chair for Extreme Environments Research sponsored by Ferring Pharmaceuticals. KSC was funded by the NERC project A-CURE (NE/P013406/1). IMB was funded by NERC (grant No. NE/R009686/1)

We thank Hamish Gordon for his advice on new particle formation schemes in the model. We acknowledge use of the Monsoon2 system, a collaborative facility supplied under the Joint Weather and Climate Research Programme, a strategic partnership between the Met Office and the Natural Environment Research Council. This work used JASMIN, the UK's collaborative data analysis environment (https://jasmin.ac.uk).

This work is part of the ASCOS and Arctic Ocean (AO) 2018 expeditions. The Swedish Polar Research Secretariat (SPRS) provided access to the icebreaker Oden and logistical support. We are grateful to the Chief Scientists Caroline Leck, Patricia Matrai and Michael Tjernström for planning and coordination of ASCOS and AO2018, to the SPRS logistical staff and to I/B Oden's Captain Mattias Peterson and his crew.

We thank the two reviewers and the community commenters for their time spent giving feedback on this paper.

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
