# Peer review of "Late summer transition from a free-tropospheric to boundary layer source of Aitken mode aerosol in the high Arctic"

_EGUsphere, 2022_

## Author Response (AR2)

Thank you to both reviewers and to community commenters Rolf Sander and Caroline Leck for the time spent reviewing our manuscript and providing feedback. Detailed responses to your comments are set out below. Text copied from comments is shown in blue with responses in black. Where we use line numbers, they refer to the line numbers in the revised manuscript unless otherwise stated.

**Response to CC1**

Following our reply to this comment ("Reply on CC1" posted on 05/12/2022), we have rephrased our description of iodine processes in the introduction on lines 56-60 to make it clear that iodine, not iodic acid, is emitted, and to add a citation to (Finkenzeller, et al. 2022). The text now reads:

*"Iodine is known to be emitted from kelp in coastal areas and iodine-containing compounds have been observed to drive NPF events near the coast of Greenland (Sipila et al., 2016; Allan et al., 2015). Of relevance to this study, Baccarini et al. (2020a) recently observed NPF driven by iodic acid (HIO3) in the pack ice region of the Arctic Ocean during the sea ice freeze-up, suggesting an iodine source from snow, ice or ocean water. Recent laboratory results elucidate the chemical pathway for the creation of iodic acid from iodine (Finkenzeller et al., 2022)."*

We have also added this text at lines 261-264:

*"E can be considered a net emission rate, which also accounts for the conversion of iodine into HIO3 (the rate is assumed to be constant). This paremeterisation does not include the HIO3 chemical formation mechanism described in Finkenzeller et al. (2022). Still, it can reproduce the observed iodic acid concentration, as shown in Figure 2 and already reported in Baccarini et al. (2020a). Hence, it serves well the scope of this work."*

**Response to RC1**

The errors in the figures make the text difficult to follow. The figures have multiple lines, sometimes overlapping so some can't be seen. I see that some effort was made to improve this by making lines transparent in figure 7 - but this was not successful. Figure 3 has lines missing completely.

Figure 3 and figure 2 were the wrong way around, so this has been corrected. Figures have now been edited and some of the simulations have been moved to the appendix (see next point) which hopefully makes them clearer.

It would have been better if fewer models had been reported on in the main body of the paper and the other models runs relegated to the appendix or supplementary material. This would make the paper much easier to read.

There are now only 5 model runs in the main text of the paper, which will hopefully make it easier to follow. Also, following the suggestions of reviewer #2, some of the simulations have been renamed so that their labels are more closely related to the model perturbation they represent, rather than naming them after the authors of papers that defined the parameterisations. E.g. simulation M10_BL in the first draft, originally named after (Metzger, et al. 2010), is now called SOA for secondary organic aerosol, which is what the (Metzger, et al. 2010) scheme uses to drive particle formation. We hope that these process-orientated names will make it easier for the reader to follow the differences between simulations.

A couple of rather more minor point, in figure 2 the x-axis says days of year when the figures are clearly day-month.

The x-axes of Figures 3 and 7 have been changed to day of year.

Also, it would be interesting to know why the model was run for 2018 when the two of observational campaigns were in 2008 and 2016. Is this because there was not computer time to run more years?

The reviewer is correct that this choice was made to save computational expense. In general, we are not seeking to explain the specific behaviour of aerosol sources during the ASCOS or ATom campaigns, since our focus was the new particle formation events during AO2018. Where we have used the ASCOS and ATom data in our analysis, we were seeking to establish whether the aerosol concentrations above the surface were within reasonable ranges in our simulations with stronger new particle formation (e.g. simulation SOA) and especially in our simulation with a prescribed free-tropospheric formation rate (SOA_PRSC). We could not use only the AO2018 data for this because only surface-level data is available from that campaign. We do not claim that our comparison between these simulations and ASCOS or ATom data leads to a robust description of aerosol sources during those campaigns, but rather that the model results warrant further attention to free-tropospheric particle formation and entrainment of particles into the Arctic boundary layer.

We have added this text (lines 158-160):

*"Simulated values of N50 (number concentration of particles larger than 50 nm diameter) over a 30 period suggest that the interannual variability of N50 in the Arctic does not account for such large differences in our simulated particle concentrations* (Carslaw and Pringle 2022)*."*

which includes a reference to a book chapter that presents output of N50 (number concentration of particles larger than 50 nm diameter) from 30 year simulations. The variability of N50 in the Arctic from these simulations suggests that the orders of magnitude difference we see between our CONTROL simulation and the ATom and ASCOS measurements cannot be explained by interannual variability alone.

**Response to RC2**

Minor Comments:

1. Figure 1 - The ASCOS line looks purple to me, not pink.

Changed "pink" to "purple" in the figure caption.

2. I found the naming of simulations to be confusing and I was constantly referencing Table 1 to follow the discussion. The problem for me is that the simulations were named primarily based on the paper from which the parameterization came. I am not familiar with these papers and I could not remember which paper added which process. Process-based simulation names would have helped, such as IA which is already used for ionic acid.

We have followed this suggestion. Simulations originally labelled "M10" for (Metzger, et al. 2010) have been renamed "SOA" for secondary organic aerosol, which is what drives particle formation in the Metzger scheme. Also, on the advice of reviewer #1 more simulations have been moved to the appendix. Hopefully, the smaller number of simulations overall in the main body will also make it easier for the reader to follow which one is which in the results section.

3. Figure 2 is referenced as Figure 3 in the text and vice versa

This has been corrected.

4. Could the authors specify how overlap indices are calculated? I am not familiar with this metric.

We have added equation 5, taken from (Pastore and Calcagnì 2019), for the calculation of the overlap index.

5. Line 375: This line leads me to believe that CONTROL only includes upper tropospheric NPF. I hadn't caught this before. It would be useful to point this out in Table 1.

The scheme in CONTROL was originally designed to target NPF processes that are most relevant for the upper troposphere, i.e. very cold temperatures and very low condensation sinks from existing aerosol. We therefore referred to CONTROL as "only including upper tropospheric NPF" because in practice it is unlikely to produce any NPF events in the boundary layer. To be more precise and therefore hopefully clearer to the reader, we have removed the reference to "only including upper tropospheric NPF" from this section and instead highlighted this detail about the default NPF scheme in section 3.1 (lines 231-234), with this text: *"The scheme creates particles mostly in the cold free and upper troposphere. Here, we also run simulations with NPF schemes that also create particles in the boundary layer (BL) and lower troposphere. We do this so that we can compare NPF driven by iodic acid, observed during AO2018, to other nucleation mechanisms known to be important in the Arctic and extra-Arctic".*

6. Line 410: Missing reference (currently "?") I believe is Igel et al 2017, doi: 10.1002/2017GL073808

The reviewer is correct about the reference and this has been fixed.

7. Figure 8: I think that the idealized N<10nm timeseries doesn't quite reflect reality. I understand it is meant to be simplified, but I was surprised by the almost total lack of variability in the late summer and surprised that an increase in the average after freeze up was not included.

This part of figure 8 has been altered so that the drawing of the line in late summer is less "flat", i.e. representing more variable concentrations, and is lower than the concentrations in the freeze up.

**Response to CC2**

**General:**

Although, the manuscript cites many very relevant articles from the field the literature survey is not fully complete; there is quite a large amount of previous observational evidence that would seem to have an essential bearing on the results and conclusions obtained and, thus, appear to merit discussion. In this respect, it seems beneficial to mention of or learn from the previous work by Tjernström and Leck and their colleagues over the last three decades over the summer Arctic pack ice area (incl. the marginal ice zone).

Their work has shown that it is not only necessary to be able to specify particle concentrations in the atmosphere and their possible sources, also we must understand the thermodynamic structure of the lower atmosphere (typically a well-mixed shallow boundary layer at the surface, only a couple

hundred meters deep, capped by a temperature inversion below the free troposphere), the dynamics of the boundary layer, and processes important in exchange between the air and ocean top layers.

It should be made clear that this contrasts with the processes in more southerly latitudes, where deep convection, could enhance mixing across the whole troposphere. Over the pack ice this does not occur other than possibly in frontal zones associated with passing weather systems.

We have added a paragraph in the introduction (starting at line 124) about the importance of studying Arctic boundary layer structure in connection to aerosol sources. New paragraph:

*"A further complicating factor that affects our understanding of summertime Arctic aerosol processes is the decoupling of the surface, where most aerosol measurements have been made in the high Arctic, and the cloud layer, where the aerosols act as CCN. Such decoupling is caused by the thermodynamic structure of the high Arctic summer boundary layer. Regimes of turbulence at the surface and in the cloud mixed layer have previously been shown to occur in configurations where the two regimes do not interact, inhibiting transport of moisture or particles vertically between the different layers (Shupe et al., 2013; Sotiropoulou et al., 2014; Brooks et al.). Decoupling of the surface in this way has implications for aerosol-cloud interactions because it implies that the aerosol sources, concentrations and size distributions at the surface may only be relevant for the cloud layer during sporadic events of mixing. This is very different from dynamics typical of lower latitudes,  where strong convection can promote mixing of heat, moisture an aerosols from the surface up to higher altitudes. Models have struggled to capture decoupling in the Arctic, showing a tendency to become coupled too often (Birch et al., 2012; Sotiropoulou et al., 2016)."*

This includes a sentence emphasising the difference between conditions in the Arctic and deep convection at lower latitudes. However, the entrainment of particles into the boundary layer does not require deep convection. Entrainment commonly occurs at the top of stratiform clouds, as a result of turbulence generated by cloud-top radiative cooling (Igel, et al. 2017). One recent study has shown that Arctic stratus clouds can sustain entrainment sufficient to maintain the cloud against loss of water by precipitation, via the mixing down of high-humidity air from the commonly observed humidity inversion (Solomon, et al. 2011).

We have also added sentences to the discussion (lines 581-585) noting how the difficulty in producing accurate model simulations of the Arctic boundary layer may complicate further study of entrainment as a source of aerosols:

*"Previous studies have shown that models struggle to simulate the thermodynamic structure of the boundary layer in the high Arctic, for example by failing to produce the decoupled conditions that are common* (Birch, et al. 2012, Sotiropoulou, et al. 2016). *Such biases could inhibit accurate predictions of free-tropospheric aerosol sources, since the entrainment of particles in the boundary layer will rely on accurate representation of the turbulent mixing created by the structure of the cloud layer."*

As such, the structure of the pack ice lower atmosphere generally limits mixing of aerosol particles in the free troposphere into the boundary layer. For this to happen it requires that aerosols sources in the free troposphere (either sourced locally or from long distant advection from lower latitudes) are brought down to the top of the boundary layer where entrainment can occur, and the only mechanism that can bring elevated plumes down to the inversion is large-scale subsidence, which is a very slow process. Entrainment is thus unlikely to be a major factor contributing to CCN (Aitken/accumulation modes) number concentrations within the lower atmosphere, and thus only

will not be a main contributor to the formation of low-level clouds (e.g., Bigg et al., 2001; Tjernström et al., 2012),

These past findings are in direct contrast to one of the main conclusions made in the present study; That particles formed outside the Arctic are the dominant source of Aitken mode particles during the sea ice melt period up to the end of August, "Particles from such remote sources, entrained into the boundary layer from the free troposphere, account for nucleation and Aitken mode particle concentrations".

Also not accounted for is the possibility that aerosols and their precursors, lofted in the deeper atmospheric upstream well mixed boundary layer over the open water, could be advected in over the pack ice on top of the shallow local boundary layer (Tjernström et al., 2012) and later be entrained into the local boundary layer through the cloud top by cloud induced mixing (Igel at at., 2017).

We do not believe that the thermodynamic structure of the Arctic boundary layer inhibits entrainment of particles, since cloud top cooling can produce turbulent mixing. Also, substantial subsidence of elevated plumes is not necessarily required to supply aerosols to the boundary layer. Aerosols are also transported (or created) just above the boundary layer, rather than at higher altitudes, and for such aerosol large-scale subsidence is not required to bring them down to the top of boundary layer. We show in Fig. 4(b) that the use of organically-mediated new particle formation in the model increases the nucleation mode aerosol concentration at the location of the *Oden* for all altitudes shown, including just above the top of the boundary layer. This is the same behaviour that is described in (Tjernström, et al. 2012) and (Igel, et al. 2017), and summarised by the commenter here. It is also described in (Solomon, et al. 2011) in the context of downward moisture transport. Our results add to this discussion by showing that our model has secondary particles above the boundary layer that can be entrained to the cloud layer or surface.

Other past observations over the pack ice (during melt and beginning of freeze-up) that should have merit discussion is the demonstration that organics found in Aitken and accumulation mode aerosols and in cloud water behaved like marine polymer gels originating from the surface microlayer on leads (open water between ice floes), due to the activity of ice microalgae, phytoplankton and perhaps, bacteria (e.g., Leck and Bigg, 2005; Bigg and Leck, 2008; Orellana et al, 2011; Hamacher-Barth et al., 2016).

We have added more references in our description of the literature surrounding Arctic primary marine organics in the introduction (lines 102-106), to include references given here and in the "Detailed" section of the comment. We have also added text in the discussion (lines 588-601) to highlight that this study did not perturb primary sources as part of the investigations into aerosol sources AO2018. The new text, given below in response to a comment in the "Detailed" section, emphasises that our model results do not rule out a primary marine source of particles in this region, and recommends that future work focuses on the balance of primary and secondary aerosol sources in the Arctic pack ice region.

Another essential bearing on the conclusions obtained not mentioned is that the thermodynamic structure of the pack ice lower atmosphere has been characterized by two predominantly near-neutrally stratified layers below the main capping inversion of the boundary layer; one in the lowest few hundred meters and one around one kilometer or slightly higher and the possibility of a recoupling and turbulent mixing between them. The reason for the two-layer boundary layer structure is likely a combination of surface-based turbulent mixing from below and cloud-top buoyancy-driven mixing from aloft (e.g., Shup et al., 2013; Brooks et al. 2017).

A description of these "decoupled" conditions is now included in lines 124-134 (given above), following the suggestion above to highlight the importance of understanding the thermodynamic structure of the Arctic boundary layer.

**Detailed:**

**Line 20-21:** "Clouds are a major control on the surface energy balance in the Arctic. Due to the low solar insolation and high albedo of sea ice in the high Arctic…" This is true but only valid over the pack ice area ca north of 80°N. Please clarify.

We have added a definition of the "high Arctic", as we called it, to be *"the pack ice region north of 80°"* (line 23).

**Line 31:** Define "high Arctic". If north of 80°, describe the sources of anthropogenic primary emissions.

The commenter is correct that we are still referring to areas north of 80°. The anthropogenic emissions we mention here refer to long-range transport of anthropogenic aerosol during the Arctic haze season. We have rephrased the sentence to make this clearer.

**Line 34:** Clarify what is meant by "thermodynamically easier".

This is referring to the air mass of the polar dome and the transport barrier it forms for air moving north from midlatitudes to the Arctic (Stohl 2006). We have added text to explain this:

*"Cold air masses sitting over ice-covered portions of the Arctic Ocean create sharp north-south temperature gradients, described as the polar dome, that act as a barrier for air moving towards the Arctic lower troposphere from the south. In the winter, several factors make it easier for air from south to penetrate the polar dome, including a more southward extent of the dome, and cooling of air during transport due to proximity to snow and ice surfaces over land."*

**Lin 36-39:** Make clear that the references to back up the statement concerning the aerosol seasonal cycles are only valid for latitudes south of the Arctic pack ice.

Most of the references used here relate to studies conducted in Svalbard, and we have now made this clear. One reference was pan-Arctic in scope, though still only includes measurements from the continental Arctic where ground stations are. However, we do not believe this means the results of these studies do not apply to the pack ice region. We have added a more recent reference, (Boyer, et al. 2023) with results from the MOSAiC campaign of a year of aerosol measurements from the pack ice region. The study shows that the seasonal cycle we describe (a peak in mass in winter/early spring and a peak in number in summer) is applicable to the pack ice region as well as Arctic sites further south, despite some differences.

Relevant papers to be added are:

Karl, M., C. Leck, F. Mashayekhy Rad, A. Bäcklund, S. Lopez-Aparicio, J. Heintzenberg, 2019, New insights in sources of the sub-micrometre aerosol at Mt. Zeppelin observatory (Spitsbergen) in the year 2015, Tellus B, 71 (1), 1-29.

We have added this reference at line 44.

Heintzenberg, J., Tunved, P., Gali, M., and Leck, C., 2017, New particle formation in the Svalbard region 2006–2015, Atmos. Chem. Phys., 17, 10, doi:10.5194/acp-2016-1073.

**Line 46:** Please add Heintzenberg et al., 2017

We have added this reference at line 52.

**Line 95:** It should be made clear to the reader that these observations are made over the high Arctic pack ice area. The break-up theory was first introduced by Leck and Bigg, 1999 followed up by Leck and Bigg, 2010, please add prior to Lawler et al., 2021.

Leck, C., and E.K. Bigg, 1999, Aerosol production over remote marine areas - A new route, Geophys. Res. Lett., 23, 3577-3581.

On line 102 We have changed "Arctic" to *"the high Arctic pack ice region"*. We have added these references as suggested, and others mentioned in the "General" section of this comment.

**Line 110:** Bulatovic et al., 2021's modeling study was set out to to explore if Aitken mode particles can act as CCN and influence the cloud properties when accumulation mode aerosols are low in number. The aerosol particle concentrations used in the simulations were chosen to cover a range of typical aerosol size distributions encountered in the summertime central Arctic during four previous campaigns in summers of 1991, 1996, 2001 and 2008 (Heintzenberg and Leck, 2012).

Based on simulated median supersaturations between 0.2 and 0.4 % ranging up to 1 %, the calculated activation diameters were as low as ~ 30 nm, suggesting that Aitken mode aerosols could be activated. Further, the authors examined the representativeness (i.e. how frequently these types of distributions occur in the observations) of the assumed conditions with low accumulation mode concentrations (i.e. lower than 20cm$^{-3}$). For two classes of Aitken mode number concentrations 100 < AIT < 200 cm$^{-3}$ and AIT < 25 cm$^{-3}$, the occurrence probability showed to be of 5% and 17% of total minutes of observations, respectively. As such there seems to be a low probability for the combination of low numbers of accumulation mode particles and high numbers of the Aitken mode particle concentration.

Please add the reported frequency of occurrence of observations supporting the simulated activation of Aitken mode particles.

We have added the reported statistics on line 119, adding:

*"…though Bulatovic et al., (2021) find that two sets of aerosol conditions under which Aitken activation is favourable in their model have real-world occurrence probabilities of 5 and 17%, raising questions about how widespread the phenomenon could be"*

**Line 110:** Please replace Vüllers et al., 2020 with Leck et al., 2020.

Leck, C., Matrai, P., Perttu A-M., and Gårdfeldt, K.: Expedition report: SWEDARTIC Arctic Ocean 2018, ISBN 978-91-519-3671-0, 2020.

We have added this report, although we have retained the reference to (Vüllers, et al. 2020) since it provides a useful summary of meteorological results and conditions during the campaign.

**Line 118:** "the long time series" relative to what?

We have made this sentence more precise by changing *"long time series"* to *"...the time series of particle concentrations and size distributions, spanning several weeks over the end of the sea ice melt period and transition to freeze-up...".*

**Line 120:** Heintzenberg et al., 2015 gives a detailed discussion on possible aerosol sources for the central Arctic pack ice area based on observations from previous expeditions in 1991, 1996, 2001 and 2008. Please add their reported results to the discussion.

Heintzenberg, J., C. Leck, and Tunved, P., 2015, Potential source regions and processes of the aerosol in the summer Arctic, Atmos. Chem. Phys., 15, 6487-6502, doi:10.5194/acp-15-6487-2015.

We have not added this reference here because this paragraph is specifically about the AO2018 campaign, so it is not clear how this study about previous campaigns should fit into the text.

**Line 228-235, Equation (3):** In the steady-state model of Baccarini et al. (2020a), nucleation of iodic acid is missing as sink. At a nucleation rate of $1/cm^3/s$ and two $HIO_3$ molecules in the activated cluster it is around 500 molecules $HIO_3$ per cubic centimeter and second. This might be small compared to the more important losses by dry deposition and condensation but could be relevant if nucleation is strong while number of pre-existing particles is low. At least it should be mentioned that loss by nucleation is neglected in Equation (3).

The uptake of gaseous $HIO_3$ to fog/cloud water is also not considered in Equation (3).

The gas is actively depleted by particle formation in the model so it doesn't need to be accounted for in this equation for the concentration due to emissions. We have attempted to make this clearer to the reader by adding to a sentence on line 268: *"…a process which also depletes the iodic acid gas concentration"*.

The sink from fog/cloud droplets is emphasised on line 264-266 and again in the discussion in lines 621-624.

**Line 249-251:** It is certainly not clear why the assumption of instantaneous homogenous distribution of $HIO_3$ throughout the boundary layer had to be made in a global 3-D model. It is very likely that $HIO_3$ is concentrated at the surface, also because its lifetime is probably less than 1 hour as it mainly depends on the condensation sink typically being in the range of 10^-4 to 10^-21/s.

Because the equation for iodic acid concentration from (Baccarini, et al. 2020), our equation 3, is given for steady state values, the iodic acid concentration in each gridbox is overwritten in each timestep. Therefore, we could not use the model's mixing or advection schemes to calculate mixing of iodic acid online in the model run. As such, a decision was needed about where to distribute the iodic acid in the boundary layer and we made the assumption that the lifetime of the iodic acid is long enough to allow mixing, given the low condensation sink from existing aerosol.

As shown in Fig. 2, our approach leads to reasonable agreement with the observed iodic acid surface concentrations, so a decision to concentrate the gas at the surface would likely have resulted in a high bias in the model. Therefore, the approach we took is sufficient to make a first test of the effect of particle formation from the gas in the model.

**Line 272:** In the XXX_SecOrg case runs, the oxidation rate of monoterpenes was reduced by 100. This is also an unjustified assumption since the oxidation rate of monoterpenes is quite accurately known, maybe with an uncertainty of 10-30 %. Would it not be more likely that gaseous semivolatile oxidation products that form in the free troposphere are entrained into the boundary layer, molecular diffusion is much more feasible than the entrainment and downward transport of particles.

Our choice to adjust the monoterpene oxidation rate was not designed to reflect uncertainty in the oxidation rate, but rather because other precursor species, with different volatilites and therefore slower oxidation rates, are missing from the model, so not a lot of precursor gas is transported to the Arctic. We have added text to emphasise this choice in lines 306-308 to make it clearer to the reader:

*"We alter the oxidation rate of monoterpene to promote transport of monoterpenes north, to account for missing species and reactions that create organic aerosol precursors. This approach allows us to test the effect of neglected organic species with oxidation rates different from monoterpenes."*

**Line 139:** Please add Leck et al., 2020.

Added.

**Line 160:** After concentrations, please add Leck et al., 2022.

Leck, C., J., Sedlar, E., Swietlicki, S., Sjögren, B.,  Brooks, S., Norris (2022) Vertical stratification of submicrometer aerosol particles measured during the high-Arctic ASCOS expedition 2008. Dataset version 1. Bolin Centre Database. https://doi.org/10.17043/oden-ascos-2008-aerosol-stratification-1

Added.

**Line 274-275:** I agree that more work is needed to assess the role of fog in controlling the frequency of iodic acid new particle formation events over the central Arctic pack ice area. However, the control of iodic acid by fog does still not explain the, during past expedition to the same area and at the time of early freeze-up (e.g., Leck and Bigg, 1999; Karl et al., 2013), simultaneous increases in particle numbers occurring in certain size ranges below 50 nm diameter. Also present were accumulation mode particles marine in origin. Stable air masses with at least 4 days or longer residence time over the ice, a surface mixed layer of 100-200m, capped by a temperature inversion and a cloud free stable layer about 1km in depth excluded a tropospheric source.

Assuming this is referring to lines 574-575 from the first draft:

We do not believe that these previous measurements – which were taken during cloud-free episodes - rule out the possibility of an entrained aerosol source – which would be driven by cloud-top mixing when clouds are present. The two sources could both exist and dominate at different times or in different places depending on local conditions. This is a very interesting open research question, since the surface and the free troposphere will have different sensitivies to the rapidly declining sea ice, and we highlight this in the discussion (lines 577-581).

We have added text in the discussion that emphasises the possibility of other, primary sources of aerosols. This was also added in response to comments in the "General" section of this comment. The new text highlights that our model results cannot rule out such sources, since we only focus on secondary sources (lines 588-601):

*"We did not consider primary sources for the Aitken mode during AO2018, such as biogenic marine particles. Although our model includes primary marine aerosol emissions, these are mostly limited to the accumulation mode. Thus, our model does not include direct marine emissions of smaller particles. Previous field studies in the Arctic pack ice region have found evidence that organics in Aitken and accumulation mode aerosols and cloud water were related to polymer gels found in the surface microlayer of sea water, suggesting a primary marine aerosol source (Bigg et al., 2001; Leck and Bigg, 2005; Bigg and Leck, 2008; Leck and Bigg, 2010; Orellana et al., 2011; Karl et al., 2013; Hamacher-Barth et al., 2016), although our earlier modelling results suggest that to account for the observed Aitken mode concentrations would require an unrealistically high surface source (Korhonen, et al. 2008). Also proposed in the literature is an atmospheric processing pathway where larger primary particles break-up to form more, smaller particles. We cannot use our model results to exclude such a source from the aerosol size distributions measured during AO2018. The balance of primary and secondary aerosol sources in this region merits further work, and would likely require improvements to existing model parameterisations (i.e. new particle formation and growth rates, primary marine emission fluxes) and the development of new parameterisations to test in the model (i.e. particle emissions from open leads independent of wind speed, and size-resolved break-up rates for primary marine particles."*

In the new text we have added a citation to a previous modelling study that found that to account for measurements of Arctic Aitken mode concentrations using a marine primary source, the high particle flux necessary cannot be reconciled with measurements of surface particle emissions in the Arctic.

It is also worth mentioning here that conditions during AO2018 were somewhat different from previous campaigns in the approximately the same season and place, with more multi-layered clouds observed that could act to promote vertical mixing relative to previous years (Vüllers, et al. 2020).

**Line 580:** Please replace "The data used here from the ASCOS campaign is available at www.ascos.se." with "The data used here from the ASCOS campaign is available on the Bolin Centre Database. https://doi.org/10.17043/oden-ascos-2008-aerosol-stratification-1"

Changed.

**Line 640:** ALL publications relating to ASCOS and AO2018 (MOCCHA, ACAS, ICE) must include the following (minimum) acknowledgment:

"This work is part of the ASCOS and Arctic Ocean (AO) 2018 expeditions. The Swedish Polar Research Secretariat (SPRS) provided access to the icebreaker Oden and logistical support. We are grateful to the Chief Scientists Caroline Leck, Patricia Matrai and Michael Tjernström for planning and coordination of ASCOS and AO2018, to the SPRS logistical staff and to I/B Oden's Captain Mattias Peterson and his crew".

Added.

**References**

Baccarini, Andrea, Linn Karlsson, Josef Dommen, Patrick Duplessis, Jutta Vüllers, Ian M. Brooks, Alfonso Saiz-Lopez, et al. 2020. "Frequent new particle formation over the high Arctic pack

ice by enhanced iodine emissions." *Nature Communications*, 12: 4924.
http://www.nature.com/articles/s41467-020-18551-0.

Birch, C. E., I. M. Brooks, M. Tjernström, M. D. Shupe, T. Mauritsen, J. Sedlar, A. P. Lock, et al. 2012.
"Modelling atmospheric structure, cloud and their response to CCN in the central Arctic:
ASCOS case studies." *Atmospheric Chemistry and Physics* 12 (7): 3419-3435.
https://www.atmos-chem-phys.net/12/3419/2012/.

Boyer, Matthew, Diego Aliaga, Jakob Boyd Pernov, Hélène Angot, Lauriane L. J. Quéléver, Lubna
Dada, Benjamin Heutte, et al. 2023. "A full year of aerosol size distribution data from the
central Arctic under an extreme positive Arctic Oscillation: insights from the Multidisciplinary
drifting Observatory for the Study of Arctic Climate (MOSAiC) expedition." *Atmospheric
Chemistry and Physics*, 1: 389-415. https://acp.copernicus.org/articles/23/389/2023/.

Carslaw, Ken S., and Kirsty Pringle. 2022. "Global aerosol properties." In *Aerosols and Climate*, by Ken
S. Carslaw, edited by Ken S. Carslaw, 101-133. Elsevier.

Finkenzeller, Henning, Siddharth Iyer, Xu-Cheng He, Mario Simon, Theodore K. Koenig, Christopher F.
Lee, Rashid Valiev, et al. 2022. "The gas-phase formation mechanism of iodic acid as an
atmospheric aerosol source." *Nature Chemistry*, 11.

Igel, Adele L., Annica M.L. Ekman, Caroline Leck, Michael Tjernström, Julien Savre, and Joseph Sedlar.
2017. "The free troposphere as a potential source of arctic boundary layer aerosol particles."
*Geophysical Research Letters.*

Korhonen, Hannele, Kenneth S. Carslaw, Dominick V. Spracklen, David A. Ridley, and Johan Ström.
2008. "A global model study of processes controlling aerosol size distributions in the Arctic
spring and summer." *Journal of Geophysical Research* (Wiley-Blackwell) 113 (D8): D08211.
http://doi.wiley.com/10.1029/2007JD009114.

Metzger, Axel, Bart Verheggen, Josef Dommen, Jonathan Duplissy, Andre S.H. Prevot, Ernest
Weingartner, Ilona Riipinen, et al. 2010. "Evidence for the role of organics in aerosol particle
formation under atmospheric conditions." *Proceedings of the National Academy of Sciences
of the United States of America*, 4: 6646-6651.

Pastore, Massimiliano, and Antonio Calcagnì. 2019. "Measuring distribution similarities between
samples: A distribution-free overlapping index." *Frontiers in Psychology*, 5.

Solomon, A., M. D. Shupe, P. O.G. Persson, and H. Morrison. 2011. "Moisture and dynamical
interactions maintaining decoupled Arctic mixed-phase stratocumulus in the presence of a
humidity inversion." *Atmospheric Chemistry and Physics.*

Sotiropoulou, Georgia, Joseph Sedlar, Richard Forbes, and Michael Tjernström. 2016. "Summer
Arctic clouds in the ECMWF forecast model: an evaluation of cloud parametrization
schemes." *Quarterly Journal of the Royal Meteorological Society* (John Wiley & Sons, Ltd)
142 (694): 387-400. http://doi.wiley.com/10.1002/qj.2658.

Stohl, A. 2006. "Characteristics of atmospheric transport into the Arctic troposphere." *Journal of
Geophysical Research*, 6: D11306. http://doi.wiley.com/10.1029/2005JD006888.

Tjernström, M., C. E. Birch, I. M. Brooks, M. D. Shupe, P. O.G. Persson, J. Sedlar, T. Mauritsen, et al.
2012. "Meteorological conditions in the central Arctic summer during the Arctic Summer
Cloud Ocean Study (ASCOS)." *Atmospheric Chemistry and Physics*, 6863-6889.

Vüllers, Jutta, Peggy Achtert, Ian Brooks, Michael Tjernström, John Prytherch, and Ryan Neely III. 2020. "Meteorological and cloud conditions during the Arctic Ocean 2018 expedition." *Atmospheric Chemistry and Physics*, 1-43.